# Bootstrap Your Own Latent
# A New Approach to Self-Supervised Learning

**Jean-Bastien Grill**[*,1] , **Florian Strub**[*,1] , **Florent Altché**[*,1] , **Corentin Tallec**[*,1] , **Pierre H. Richemond**[*,1,2]
**Elena Buchatskaya**[1] , **Carl Doersch**[1] , **Bernardo Avila Pires**[1] , **Zhaohan Daniel Guo**[1]
**Mohammad Gheshlaghi Azar**[1], **Bilal Piot**[1], **Koray Kavukcuoglu**[1] , **Rémi Munos**[1] , **Michal Valko**[1]

[1]DeepMind     [2]Imperial College

`[jbgrill,fstrub,altche,corentint,richemond]@google.com`

## Abstract

We introduce **B**ootstrap **Y**our **O**wn **L**atent (BYOL), a new approach to self-supervised image representation learning. BYOL relies on two neural networks, referred to as *online* and *target* networks, that interact and learn from each other. From an augmented view of an image, we train the online network to predict the target network representation of the same image under a different augmented view. At the same time, we update the target network with a slow-moving average of the online network. While state-of-the art methods rely on negative pairs, BYOL achieves a new state of the art *without them*. BYOL reaches 74.3% top-1 classification accuracy on ImageNet using a linear evaluation with a ResNet-50 architecture and 79.6% with a larger ResNet. We show that BYOL performs on par or better than the current state of the art on both transfer and semi-supervised benchmarks. Our implementation and pretrained models are given on GitHub.[3]

## 1   Introduction

Learning good image representations is a key challenge in computer vision [1, 2, 3] as it allows for efficient training on downstream tasks [4, 5, 6, 7]. Many different training approaches have been proposed to learn such representations, usually relying on visual pretext tasks. Among them, state-of-the-art contrastive methods [8, 9, 10, 11, 12] are trained by reducing the distance between representations of different augmented views of the same image ('positive pairs'), and increasing the distance between representations of augmented views from different images ('negative pairs'). These methods need careful treatment of negative pairs [13] by either relying on large batch sizes [8, 12], memory banks [9] or customized mining strategies [14, 15] to retrieve the negative pairs. In addition, their performance critically depends on the choice of image augmentations [34, 11, 8, 12].

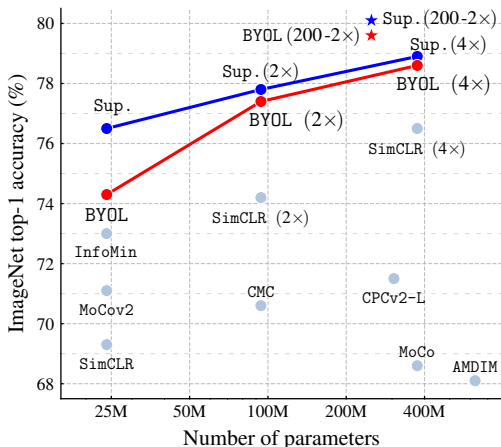

Figure 1: Performance of BYOL on ImageNet (linear evaluation) using ResNet-50 and our best architecture ResNet-200 (2×), compared to other unsupervised and supervised (Sup.) baselines [8].

---

[*]Equal contribution; the order of first authors was randomly selected.

[3]https://github.com/deepmind/deepmind-research/tree/master/byol

In this paper, we introduce **B**ootstrap **Y**our **O**wn **L**atent (BYOL), a new algorithm for self-supervised learning of image representations. BYOL achieves higher performance than state-of-the-art contrastive methods without using negative pairs. It iteratively bootstraps[4] the outputs of a network to serve as targets for an enhanced representation. Moreover, BYOL is more robust to the choice of image augmentations than contrastive methods; we suspect that not relying on negative pairs is one of the leading reasons for its improved robustness. While previous methods based on bootstrapping have used pseudo-labels [16], cluster indices [17] or a handful of labels [18, 19, 20], we propose to *directly bootstrap the representations*. In particular, BYOL uses two neural networks, referred to as online and target networks, that interact and learn from each other. Starting from an augmented view of an image, BYOL trains its online network to predict the target network's representation of another augmented view of the same image. While this objective admits collapsed solutions, e.g., outputting the same vector for all images, we empirically show that BYOL does not converge to such solutions. We hypothesize (Section 3.2) that the combination of (i) the addition of a predictor to the online network and (ii) the use of a moving average of online parameters, as the target network encourages encoding more and more information in the online projection and avoids collapsed solutions.

We evaluate the representation learned by BYOL on ImageNet [21] and other vision benchmarks using ResNet architectures [22]. Under the linear evaluation protocol on ImageNet, consisting in training a linear classifier on top of the frozen representation, BYOL reaches 74.3% top-1 accuracy with a standard ResNet-50 and 79.6% top-1 accuracy with a larger ResNet (Figure 1). In the semi-supervised and transfer settings on ImageNet, we obtain results on par or superior to the current state of the art. Our contributions are: (*i*) We introduce BYOL, a self-supervised representation learning method (Section 3) which achieves state-of-the-art results under the linear evaluation protocol on ImageNet without using negative pairs. (*ii*) We show that our learned representation outperforms the state of the art on semi-supervised and transfer benchmarks (Section 4). (*iii*) We show that BYOL is more resilient to changes in the batch size and in the set of image augmentations compared to its contrastive counterparts (Section 5). In particular, BYOL suffers a much smaller performance drop than SimCLR, a strong contrastive baseline, when only using random crops as image augmentations.

## 2   Related work

Most unsupervised methods for representation learning can be categorized as either generative or discriminative [23, 8]. Generative approaches to representation learning build a distribution over data and latent embedding and use the learned embeddings as image representations. Many of these approaches rely either on auto-encoding of images [24, 25, 26] or on adversarial learning [27], jointly modelling data and representation [28, 29, 30, 31]. Generative methods typically operate directly in pixel space. This however is computationally expensive, and the high level of detail required for image generation may not be necessary for representation learning.

Among discriminative methods, contrastive methods [9, 10, 32, 33, 34, 11, 35, 36] currently achieve state-of-the-art performance in self-supervised learning [37, 8, 38, 12]. Contrastive approaches avoid a costly generation step in pixel space by bringing representation of different views of the same image closer ('positive pairs'), and spreading representations of views from different images ('negative pairs') apart [39, 40]. Contrastive methods often require comparing each example with many other examples to work well [9, 8] prompting the question of whether using negative pairs is necessary.

DeepCluster [17] partially answers this question. It uses bootstrapping on previous versions of its representation to produce targets for the next representation; it clusters data points using the prior representation, and uses the cluster index of each sample as a classification target for the new representation. While avoiding the use of negative pairs, this requires a costly clustering phase and specific precautions to avoid collapsing to trivial solutions.

Some self-supervised methods are not contrastive but rely on using auxiliary handcrafted prediction tasks to learn their representation. In particular, relative patch prediction [23, 40], colorizing grayscale images [41, 42], image inpainting [43], image jigsaw puzzle [44], image super-resolution [45], and geometric transformations [46, 47] have been shown to be useful. Yet, even with suitable architectures [48], these methods are being outperformed by contrastive methods [37, 8, 12].

Our approach has some similarities with *Predictions of Bootstrapped Latents* (PBL, [49]), a self-supervised representation learning technique for reinforcement learning (RL). PBL jointly trains the agent's history representation and an encoding of future observations. The observation encoding is used as a target to train the agent's representation, and the agent's representation as a target to train the observation encoding. Unlike PBL, BYOL uses a slow-moving average of its representation to provide its targets, and does not require a second network.

The idea of using a slow-moving average target network to produce stable targets for the online network was inspired by deep RL [50, 51, 52, 53]. Target networks stabilize the bootstrapping updates provided by the Bellman equation, making them appealing to stabilize the bootstrap mechanism in BYOL. While most RL methods use fixed target networks, BYOL uses a weighted moving average of previous networks (as in [54]) in order to provide smoother changes in the target representation.

In the semi-supervised setting [55, 56], an unsupervised loss is combined with a classification loss over a handful of labels to ground the training [19, 20, 57, 58, 59, 60, 61, 62]. Among these methods, *mean teacher* (MT) [20] also uses a slow-moving average network, called *teacher*, to produce targets for an online network, called *student*. An $\ell_2$ consistency loss between the softmax predictions of the teacher and the student is added to the classification loss. While [20] demonstrates the effectiveness of MT in the semi-supervised learning case, in Section 5 we show that a similar approach collapses when removing the classification loss. In contrast, BYOL introduces an additional predictor on top of the online network, which prevents collapse.

Finally, in self-supervised learning, MoCo [9] uses a slow-moving average network (*momentum encoder*) to maintain consistent representations of negative pairs drawn from a memory bank. Instead, BYOL uses a moving average network to produce prediction targets as a means of stabilizing the bootstrap step. We show in Section 5 that this mere stabilizing effect can also improve existing contrastive methods.

# 3   Method

We start by motivating our method before explaining its details in Section 3.1. Many successful self-supervised learning approaches build upon the cross-view prediction framework introduced in [63]. Typically, these approaches learn representations by predicting different views (e.g., different random crops) of the same image from one another. Many such approaches cast the prediction problem directly in representation space: the representation of an augmented view of an image should be predictive of the representation of another augmented view of the same image. However, predicting directly in representation space can lead to collapsed representations: for instance, a representation that is constant across views is always fully predictive of itself. Contrastive methods circumvent this problem by reformulating the prediction problem into one of discrimination: from the representation of an augmented view, they learn to discriminate between the representation of another augmented view of the same image, and the representations of augmented views of different images. In the vast majority of cases, this prevents the training from finding collapsed representations. Yet, this discriminative approach typically requires comparing each representation of an augmented view with many negative examples, to find ones sufficiently close to make the discrimination task challenging. In this work, we thus tasked ourselves to find out whether these negative examples are indispensable to prevent collapsing while preserving high performance.

To prevent collapse, a straightforward solution is to use a fixed randomly initialized network to produce the targets for our predictions. While avoiding collapse, it empirically does not result in very good representations. Nonetheless, it is interesting to note that the representation obtained using this procedure can already be much better than the initial fixed representation. In our ablation study (Section 5), we apply this procedure by predicting a fixed randomly initialized network and achieve $18.8\%$ top-1 accuracy (Table 5a) on the linear evaluation protocol on ImageNet, whereas the randomly initialized network only achieves $1.4\%$ by itself. This experimental finding is the core motivation for BYOL: from a given representation, referred to as *target*, we can train a new, potentially enhanced representation, referred to as *online*, by predicting the target representation. From there, we can expect to build a sequence of representations of increasing quality by iterating this procedure, using subsequent online networks as new target networks for further training. In practice, BYOL generalizes this bootstrapping procedure by iteratively refining its representation, but using a slowly moving exponential average of the online network as the target network instead of fixed checkpoints.

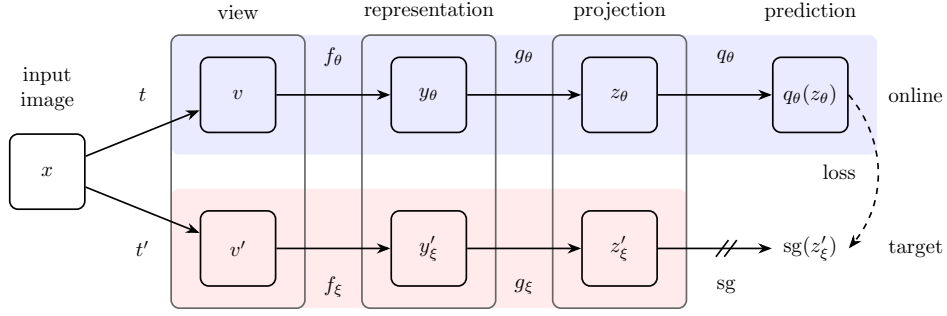

Figure 2: BYOL's architecture. BYOL minimizes a similarity loss between $q_\theta(z_\theta)$ and $\mathrm{sg}(z'_\xi)$, where $\theta$ are the trained weights, $\xi$ are an exponential moving average of $\theta$ and sg means stop-gradient. At the end of training, everything but $f_\theta$ is discarded, and $y_\theta$ is used as the image representation.

### 3.1 Description of BYOL

BYOL's goal is to learn a representation $y_\theta$ which can then be used for downstream tasks. As described previously, BYOL uses two neural networks to learn: the *online* and *target* networks. The online network is defined by a set of weights $\theta$ and is comprised of three stages: an *encoder* $f_\theta$, a *projector* $g_\theta$ and a *predictor* $q_\theta$, as shown in Figure 2 and Figure 8. The target network has the same architecture as the online network, but uses a different set of weights $\xi$. The target network provides the regression targets to train the online network, and its parameters $\xi$ are an exponential moving average of the online parameters $\theta$ [54]. More precisely, given a target decay rate $\tau \in [0, 1]$, after each training step we perform the following update,

$$\xi \leftarrow \tau \xi + (1 - \tau)\theta. \tag{1}$$

Given a set of images $\mathcal{D}$, an image $x \sim \mathcal{D}$ sampled uniformly from $\mathcal{D}$, and two distributions of image augmentations $\mathcal{T}$ and $\mathcal{T}'$, BYOL produces two augmented views $v \triangleq t(x)$ and $v' \triangleq t'(x)$ from $x$ by applying respectively image augmentations $t \sim \mathcal{T}$ and $t' \sim \mathcal{T}'$. From the first augmented view $v$, the online network outputs a *representation* $y_\theta \triangleq f_\theta(v)$ and a projection $z_\theta \triangleq g_\theta(y)$. The target network outputs $y'_\xi \triangleq f_\xi(v')$ and the *target projection* $z'_\xi \triangleq g_\xi(y')$ from the second augmented view $v'$. We then output a *prediction* $q_\theta(z_\theta)$ of $z'_\xi$ and $\ell_2$-normalize both $q_\theta(z_\theta)$ and $z'_\xi$ to $\overline{q_\theta}(z_\theta) \triangleq q_\theta(z_\theta)/\|q_\theta(z_\theta)\|_2$ and $\overline{z}'_\xi \triangleq z'_\xi/\|z'_\xi\|_2$. Note that this predictor is only applied to the online branch, making the architecture asymmetric between the online and target pipeline. Finally we define the following mean squared error between the normalized predictions and target projections,[5]

$$\mathcal{L}_{\theta,\xi} \triangleq \left\| \overline{q_\theta}(z_\theta) - \overline{z}'_\xi \right\|_2^2 = 2 - 2 \cdot \frac{\langle q_\theta(z_\theta), z'_\xi \rangle}{\left\| q_\theta(z_\theta) \right\|_2 \cdot \left\| z'_\xi \right\|_2}. \tag{2}$$

We symmetrize the loss $\mathcal{L}_{\theta,\xi}$ in Eq. 2 by separately feeding $v'$ to the online network and $v$ to the target network to compute $\widetilde{\mathcal{L}}_{\theta,\xi}$. At each training step, we perform a stochastic optimization step to minimize $\mathcal{L}_{\theta,\xi}^{\mathrm{BYOL}} = \mathcal{L}_{\theta,\xi} + \widetilde{\mathcal{L}}_{\theta,\xi}$ with respect to $\theta$ only, but *not* $\xi$, as depicted by the stop-gradient in Figure 2. BYOL's dynamics are summarized as

$$\theta \leftarrow \mathrm{optimizer}\left(\theta, \nabla_\theta \mathcal{L}_{\theta,\xi}^{\mathrm{BYOL}}, \eta\right) \quad \text{and} \quad \xi \leftarrow \tau \xi + (1 - \tau)\theta,$$

where $\mathrm{optimizer}$ is an optimizer and $\eta$ is a learning rate. At the end of training, we only keep the encoder $f_\theta$; as in [9]. When comparing to other methods, we consider the number of inference-time weights only in the final representation $f_\theta$. The architecture, hyper-parameters and training details are specified in Appendix A, the full training procedure is summarized in Appendix B, and python pseudo-code based on the libraries JAX [64] and Haiku [65] is provided in in Appendix J.

### 3.2 Intuitions on BYOL's behavior

As BYOL does not use an explicit term to prevent collapse (such as negative examples [10]) while minimizing $\mathcal{L}_{\theta,\xi}^{\mathrm{BYOL}}$ with respect to $\theta$, it may seem that BYOL should converge to a minimum of this loss

| Method | Top-1 | Top-5 |
|---|---|---|
| Local Agg. | 60.2 | - |
| PIRL [35] | 63.6 | - |
| CPC v2 [32] | 63.8 | 85.3 |
| CMC [11] | 66.2 | 87.0 |
| SimCLR [8] | 69.3 | 89.0 |
| MoCo v2 [37] | 71.1 | - |
| InfoMin Aug. [12] | 73.0 | 91.1 |
| BYOL (ours) | **74.3** | **91.6** |

(a) ResNet-50 encoder.

| Method | Architecture | Param. | Top-1 | Top-5 |
|---|---|---|---|---|
| SimCLR [8] | ResNet-50 ($2\times$) | 94M | 74.2 | 92.0 |
| CMC [11] | ResNet-50 ($2\times$) | 94M | 70.6 | 89.7 |
| BYOL (ours) | ResNet-50 ($2\times$) | 94M | 77.4 | 93.6 |
| CPC v2 [32] | ResNet-161 | 305M | 71.5 | 90.1 |
| MoCo [9] | ResNet-50 ($4\times$) | 375M | 68.6 | - |
| SimCLR [8] | ResNet-50 ($4\times$) | 375M | 76.5 | 93.2 |
| BYOL (ours) | ResNet-50 ($4\times$) | 375M | 78.6 | 94.2 |
| BYOL (ours) | ResNet-200 ($2\times$) | 250M | **79.6** | **94.8** |

(b) Other ResNet encoder architectures.

Table 1: Top-1 and top-5 accuracies (in %) under linear evaluation on ImageNet.

with respect to $(\theta, \xi)$ (*e.g.*, a collapsed constant representation). However BYOL's target parameters $\xi$ updates are **not** in the direction of $\nabla_\xi \mathcal{L}_{\theta,\xi}^{\text{BYOL}}$. More generally, we hypothesize that there is no loss $L_{\theta,\xi}$ such that BYOL's dynamics is a gradient descent on $L$ jointly over $\theta, \xi$. This is similar to GANs [66], where there is no loss that is jointly minimized w.r.t. both the discriminator and generator parameters. There is therefore no *a priori* reason why BYOL's parameters would converge to a minimum of $\mathcal{L}_{\theta,\xi}^{\text{BYOL}}$. While BYOL's dynamics still admit undesirable equilibria, we did not observe convergence to such equilibria in our experiments. In addition, when assuming BYOL's predictor to be optimal[6] i.e.,

$$q_\theta = q^\star \text{ with } q^\star \triangleq \arg\min_q \mathbb{E}\left[\left\|q(z_\theta) - z'_\xi\right\|_2^2\right], \quad \text{where} \quad q^\star(z_\theta) = \mathbb{E}\left[z'_\xi | z_\theta\right], \qquad (3)$$

we hypothesize that the undesirable equilibria are unstable. Indeed, in this optimal predictor case, BYOL's updates on $\theta$ follow in expectation the gradient of the expected conditional variance (see Appendix I for details), we note $z'_{\xi,i}$ the $i$-th feature of $z'_\xi$, then

$$\nabla_\theta \mathbb{E}\left[\left\|q^\star(z_\theta) - z'_\xi\right\|_2^2\right] = \nabla_\theta \mathbb{E}\left[\left\|\mathbb{E}\left[z'_\xi | z_\theta\right] - z'_\xi\right\|_2^2\right] = \nabla_\theta \mathbb{E}\left[\sum_i \text{Var}(z'_{\xi,i} | z_\theta)\right], \qquad (4)$$

Note that for any random variables $X$, $Y$, and $Z$, $\text{Var}(X|Y,Z) \leq \text{Var}(X|Y)$. Let $X$ be the target projection, $Y$ the current online projection, and $Z$ an additional variability on top of the online projection induced by stochasticities in the training dynamics: purely discarding information from the online projection cannot decrease the conditional variance.

In particular, BYOL avoids constant features in $z_\theta$ as, for any constant $c$ and random variables $z_\theta$ and $z'_\xi$, $\text{Var}(z'_\xi | z_\theta) \leq \text{Var}(z'_\xi | c)$; hence our hypothesis on these collapsed constant equilibria being unstable. Interestingly, if we were to minimize $\mathbb{E}[\sum_i \text{Var}(z'_{\xi,i} | z_\theta)]$ with respect to $\xi$, we would get a collapsed $z'_\xi$ as the variance is minimized for a constant $z'_\xi$. Instead, BYOL makes $\xi$ closer to $\theta$, incorporating sources of variability captured by the online projection into the target projection.

Furthermore, notice that performing a hard-copy of the online parameters $\theta$ into the target parameters $\xi$ would be enough to propagate new sources of variability. However, sudden changes in the target network might break the assumption of an optimal predictor, in which case BYOL's loss is not guaranteed to be close to the conditional variance. We hypothesize that the main role of BYOL's moving-averaged target network is to ensure the near-optimality of the predictor over training; Section 5 and Appendix J provide some empirical support of this interpretation.

| Method | Top-1 | | Top-5 | |
|---|---|---|---|---|
| | 1% | 10% | 1% | 10% |
| Supervised [77] | 25.4 | 56.4 | 48.4 | 80.4 |
| InstDisc | - | - | 39.2 | 77.4 |
| PIRL [35] | - | - | 57.2 | 83.8 |
| SimCLR [8] | 48.3 | 65.6 | 75.5 | 87.8 |
| BYOL (ours) | **53.2** | **68.8** | **78.4** | **89.0** |

(a) ResNet-50 encoder.

| Method | Architecture | Param. | Top-1 | | Top-5 | |
|---|---|---|---|---|---|---|
| | | | 1% | 10% | 1% | 10% |
| CPC v2 [32] | ResNet-161 | 305M | - | - | 77.9 | 91.2 |
| SimCLR [8] | ResNet-50 ($2\times$) | 94M | 58.5 | 71.7 | 83.0 | 91.2 |
| BYOL (ours) | ResNet-50 ($2\times$) | 94M | 62.2 | 73.5 | 84.1 | 91.7 |
| SimCLR [8] | ResNet-50 ($4\times$) | 375M | 63.0 | 74.4 | 85.8 | 92.6 |
| BYOL (ours) | ResNet-50 ($4\times$) | 375M | 69.1 | 75.7 | 87.9 | 92.5 |
| BYOL (ours) | ResNet-200 ($2\times$) | 250M | **71.2** | **77.7** | **89.5** | **93.7** |

(b) Other ResNet encoder architectures.

Table 2: Semi-supervised training with a fraction of ImageNet labels.

| Method | Food101 | CIFAR10 | CIFAR100 | Birdsnap | SUN397 | Cars | Aircraft | VOC2007 | DTD | Pets | Caltech-101 | Flowers |
|---|---|---|---|---|---|---|---|---|---|---|---|---|
| *Linear evaluation:* | | | | | | | | | | | | |
| BYOL (ours) | **75.3** | 91.3 | **78.4** | **57.2** | **62.2** | **67.8** | 60.6 | 82.5 | 75.5 | 90.4 | 94.2 | **96.1** |
| SimCLR (repro) | 72.8 | 90.5 | 74.4 | 42.4 | 60.6 | 49.3 | 49.8 | 81.4 | **75.7** | 84.6 | 89.3 | 92.6 |
| SimCLR [8] | 68.4 | 90.6 | 71.6 | 37.4 | 58.8 | 50.3 | 50.3 | 80.5 | 74.5 | 83.6 | 90.3 | 91.2 |
| Supervised-IN [8] | 72.3 | **93.6** | 78.3 | 53.7 | 61.9 | 66.7 | **61.0** | **82.8** | 74.9 | **91.5** | **94.5** | 94.7 |
| *Fine-tuned:* | | | | | | | | | | | | |
| BYOL (ours) | **88.5** | **97.8** | 86.1 | **76.3** | 63.7 | 91.6 | **88.1** | 85.4 | **76.2** | 91.7 | **93.8** | 97.0 |
| SimCLR (repro) | 87.5 | 97.4 | 85.3 | 75.0 | 63.9 | 91.4 | 87.6 | 84.5 | 75.4 | 89.4 | 91.7 | 96.6 |
| SimCLR [8] | 88.2 | 97.7 | 85.9 | 75.9 | 63.5 | 91.3 | 88.1 | 84.1 | 73.2 | 89.2 | 92.1 | 97.0 |
| Supervised-IN [8] | 88.3 | 97.5 | **86.4** | 75.8 | **64.3** | **92.1** | 86.0 | 85.0 | 74.6 | **92.1** | 93.3 | **97.6** |
| Random init [8] | 86.9 | 95.9 | 80.2 | 76.1 | 53.6 | 91.4 | 85.9 | 67.3 | 64.8 | 81.5 | 72.6 | 92.0 |

Table 3: Transfer learning results from ImageNet (IN) with the standard ResNet-50 architecture.

## 4 Experimental evaluation

We assess the performance of BYOL's representation after self-supervised pretraining on the training set of the ImageNet ILSVRC-2012 dataset [21]. We first evaluate it on ImageNet (IN) in both linear evaluation and semi-supervised setups. We then measure its transfer capabilities on other datasets and tasks, including classification, segmentation, object detection and depth estimation. For comparison, we also report scores for a representation trained using labels from the `train` ImageNet subset, referred to as Supervised-IN. In Appendix F, we assess the generality of BYOL by pretraining a representation on the Places365-Standard dataset [73] before reproducing this evaluation protocol.

**Linear evaluation on ImageNet** We first evaluate BYOL's representation by training a linear classifier on top of the frozen representation, following the procedure described in [48, 74, 41, 10, 8], and appendix D.1; we report top-1 and top-5 accuracies in % on the `test` set in Table 1. With a standard ResNet-50 ($\times$1) BYOL obtains 74.3% top-1 accuracy (91.6% top-5 accuracy), which is a 1.3% (resp. 0.5%) improvement over the previous self-supervised state of the art [12]. This tightens the gap with respect to the supervised baseline of [8], 76.5%, but is still significantly below the stronger supervised baseline of [75], 78.9%. With deeper and wider architectures, BYOL consistently outperforms the previous state of the art (Appendix D.2), and obtains a best performance of 79.6% top-1 accuracy, ranking higher than previous self-supervised approaches. On a ResNet-50 ($4\times$) BYOL achieves 78.6%, similar to the 78.9% of the best supervised baseline in [8] for the same architecture.

**Semi-supervised training on ImageNet** Next, we evaluate the performance obtained when fine-tuning BYOL's representation on a classification task with a small subset of ImageNet's `train` set, this time using label information. We follow the semi-supervised protocol of [74, 76, 8, 32] detailed in Appendix D.1, and use the same fixed splits of respectively 1% and 10% of ImageNet labeled training data as in [8]. We report both top-1 and top-5 accuracies on the `test` set in Table 2. BYOL consistently outperforms previous approaches across a wide range of architectures. Additionally, as detailed in Appendix D.1, BYOL reaches 77.7% top-1 accuracy with ResNet-50 when fine-tuning over 100% of ImageNet labels.

**Transfer to other classification tasks** We evaluate our representation on other classification datasets to assess whether the features learned on ImageNet (IN) are generic and thus useful across image domains, or if they are ImageNet-specific. We perform linear evaluation and fine-tuning on the same set of classification tasks used in [8, 74], and carefully follow their evaluation protocol, as detailed in Appendix E. Performance is reported using standard metrics for each benchmark, and results are provided on a held-out `test` set after hyperparameter selection on a validation set. We report results in Table 3, both for linear evaluation and fine-tuning. BYOL outperforms SimCLR on all benchmarks and the Supervised-IN baseline on 7 of the 12 benchmarks, providing only slightly worse performance on the 5 remaining benchmarks. BYOL's representation can transfer to small images, e.g., CIFAR [78], landscapes, e.g., SUN397 [79] or VOC2007 [80], and textures, e.g., DTD [81].

**Transfer to other vision tasks** We evaluate our representation on different tasks relevant to computer vision practitioners, namely semantic segmentation, object detection and depth estimation. With this evaluation, we assess whether BYOL's representation generalizes beyond classification tasks.

We first evaluate BYOL on the VOC2012 semantic segmentation task as detailed in Appendix E.4, where the goal is to classify each pixel in the image [7]. We report the results in Table 4a. BYOL outperforms both the Supervised-IN baseline (+1.9 mIoU) and SimCLR (+1.1 mIoU).

Similarly, we evaluate on object detection by reproducing the setup in [9] using a Faster R-CNN architecture [82], as detailed in Appendix E.5. We fine-tune on `trainval2007` and report results on `test2007` using the standard $AP_{50}$ metric; BYOL is significantly better than the Supervised-IN baseline ($+3.1$ $AP_{50}$) and SimCLR ($+2.3$ $AP_{50}$).

Finally, we evaluate on depth estimation on the NYU v2 dataset, where the depth map of a scene is estimated given a single RGB image. Depth prediction measures how well a network represents geometry, and how well that information can be localized to pixel accuracy [40]. The setup is based on [83] and detailed in Appendix E.6. We evaluate on the commonly used `test` subset of $654$ images and report results using several common metrics in Table 4b: relative (rel) error, root mean squared (rms) error, and the percent of pixels (pct) where the error, $\max(d_{gt}/d_p, d_p/d_{gt})$, is below $1.25^n$ thresholds where $d_p$ is the predicted depth and $d_{gt}$ is the ground truth depth [40]. BYOL is better or on par with other methods for each metric. For instance, the challenging pct.$<1.25$ measure is respectively improved by $+3.5$ points and $+1.3$ points compared to supervised and SimCLR baselines.

| Method | $AP_{50}$ | mIoU |
|---|---|---|
| Supervised-IN [9] | 74.4 | 74.4 |
| MoCo [9] | 74.9 | 72.5 |
| SimCLR (repro) | 75.2 | 75.2 |
| BYOL (ours) | **77.5** | **76.3** |

(a) Transfer results in semantic segmentation and object detection.

| Method | Higher better | | | Lower better | |
| | pct.$<1.25$ | pct.$<1.25^2$ | pct.$<1.25^3$ | rms | rel |
|---|---|---|---|---|---|
| Supervised-IN [83] | 81.1 | 95.3 | 98.8 | 0.573 | **0.127** |
| SimCLR (repro) | 83.3 | 96.5 | 99.1 | 0.557 | 0.134 |
| BYOL (ours) | **84.6** | **96.7** | **99.1** | **0.541** | 0.129 |

(b) Transfer results on NYU v2 depth estimation.

Table 4: Results on transferring BYOL's representation to other vision tasks.

# 5 Building intuitions with ablations

We present ablations on BYOL to give an intuition of its behavior and performance. For reproducibility, we run each configuration of parameters over three seeds, and report the average performance. We also report the half difference between the best and worst runs when it is larger than $0.25$. Although previous works perform ablations at $100$ epochs [8, 12], we notice that relative improvements at $100$ epochs do not always hold over longer training. For this reason, we run ablations over $300$ epochs on $64$ TPU v3 cores, which yields consistent results compared to our baseline training of $1000$ epochs. For all the experiments in this section, we set the initial learning rate to $0.3$ with batch size $4096$, the weight decay to $10^{-6}$ as in SimCLR [8] and the base target decay rate $\tau_{\text{base}}$ to $0.99$. In this section we report results in top-1 accuracy on ImageNet under the linear evaluation protocol as in Appendix D.1.

**Batch size**   Among contrastive methods, the ones that draw negative examples from the minibatch suffer performance drops when their batch size is reduced. BYOL does not use negative examples and we expect it to be more robust to smaller batch sizes. To empirically verify this hypothesis, we train both BYOL and SimCLR using different batch sizes from $128$ to $4096$. To avoid re-tuning other hyperparameters, we average gradients over $N$ consecutive steps before updating the online network when reducing the batch size by a factor $N$. The target network is updated once every $N$ steps, after the update of the online network; we accumulate the $N$-steps in parallel in our runs. As shown in Figure 3a, the performance of SimCLR rapidly deteriorates with batch size, likely due to the decrease in the number of negative examples. In contrast, the performance of BYOL remains stable over a wide range of batch sizes from $256$ to $4096$, and only drops for smaller values due to batch normalization layers in the encoder.[7]

**Image augmentations**   Contrastive methods are sensitive to the choice of image augmentations. For instance, SimCLR does not work well when removing color distortion from its image augmentations. As an explanation, SimCLR shows that crops of the same image mostly share their color histograms. At the same time, color histograms vary across images. Therefore, when a contrastive task only relies on random crops as image augmentations, it can be mostly solved by focusing on color histograms alone. As a result the representation is not incentivized to retain information beyond color histograms. To prevent that, SimCLR adds color distortion to its set of image augmentations. Instead, BYOL is incentivized to keep any information captured by the target representation into its online network, to

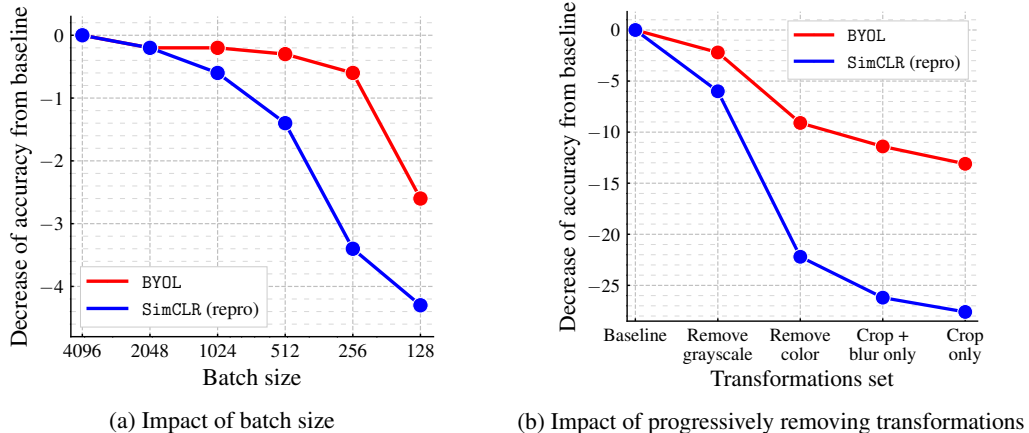

(a) Impact of batch size

(b) Impact of progressively removing transformations

Figure 3: Decrease in top-1 accuracy (in % points) of BYOL and our own reproduction of SimCLR at 300 epochs, under linear evaluation on ImageNet.

improve its predictions. Therefore, even if augmented views of a same image share the same color histogram, BYOL is still incentivized to retain additional features in its representation. For that reason, we believe that BYOL is more robust to the choice of image augmentations than contrastive methods.

Results presented in Figure 3b support this hypothesis: the performance of BYOL is much less affected than the performance of SimCLR when removing color distortions from the set of image augmentations ($-9.1$ accuracy points for BYOL, $-22.2$ accuracy points for SimCLR). When image augmentations are reduced to mere random crops, BYOL still displays good performance (59.4%, *i.e.* $-13.1$ points from 72.5% ), while SimCLR loses more than a third of its performance (40.3%, *i.e.* $-27.6$ points from 67.9%). We report additional ablations in Appendix G.3.

**Bootstrapping** BYOL uses the projected representation of a target network, whose weights are an exponential moving average of the weights of the online network, as target for its predictions. This way, the weights of the target network represent a delayed and more stable version of the weights of the online network. When the target decay rate is 1, the target network is never updated, and remains at a constant value corresponding to its initialization. When the target decay rate is 0, the target network is instantaneously updated to the online network at each step. There is a trade-off between updating the targets too often and updating them too slowly, as illustrated in Table 5a. Instantaneously updating the target network ($\tau = 0$) destabilizes training, yielding very poor performance while never updating the target ($\tau = 1$) makes the training stable but prevents iterative improvement, ending with low-quality final representation. All values of the decay rate between 0.9 and 0.999 yield performance above 68.4% top-1 accuracy at 300 epochs.

| Target | $\tau_{\text{base}}$ | Top-1 |
|---|---|---|
| Constant random network | 1 | $18.8_{\pm 0.7}$ |
| Moving average of online | 0.999 | 69.8 |
| Moving average of online | 0.99 | **72.5** |
| Moving average of online | 0.9 | 68.4 |
| Stop gradient of online[†] | 0 | 0.3 |

(a) Results for different target modes. [†]In the *stop gradient of online*, $\tau = \tau_{\text{base}} = 0$ is kept constant throughout training.

| Method | Predictor | Target network | $\beta$ | Top-1 |
|---|---|---|---|---|
| BYOL | ✓ | ✓ | 0 | **72.5** |
| — | ✓ | ✓ | 1 | 70.9 |
| — | | ✓ | 1 | 70.7 |
| SimCLR | | | 1 | 69.4 |
| — | ✓ | | 1 | 69.1 |
| — | ✓ | | 0 | 0.3 |
| — | | ✓ | 0 | 0.2 |
| — | | | 0 | 0.1 |

(b) Intermediate variants between BYOL and SimCLR.

Table 5: Ablations with top-1 accuracy (in %) at 300 epochs under linear evaluation on ImageNet.

**Ablation to contrastive methods** In this subsection, we recast SimCLR and BYOL using the same formalism to better understand where the improvement of BYOL over SimCLR comes from. Let us consider the following objective that extends the InfoNCE objective [10, 84] (see Appendix G.4),

$$\text{InfoNCE}_{\theta}^{\alpha,\beta} \triangleq \frac{2}{B}\sum_{i=1}^{B}S_{\theta}(v_i, v_i') - \beta \cdot \frac{2\alpha}{B}\sum_{i=1}^{B}\ln\left(\sum_{j\neq i}\exp\frac{S_{\theta}(v_i, v_j)}{\alpha} + \sum_{j}\exp\frac{S_{\theta}(v_i, v_j')}{\alpha}\right),$$

where $\alpha > 0$ is a fixed temperature, $\beta \in [0,1]$ a weighting coefficient, $B$ the batch size, $v$ and $v'$ are batches of augmented views where for any batch index $i$, $v_i$ and $v_i'$ are augmented views from the same image; the real-valued function $S_\theta$ quantifies pairwise similarity between augmented views. For any augmented view $u$ we denote $z_\theta(u) \triangleq f_\theta(g_\theta(u))$ and $z_\xi(u) \triangleq f_\xi(g_\xi(u))$. For given $\phi$ and $\psi$, we consider the normalized dot product

$$S_\theta(u_1, u_2) \triangleq \frac{\langle\phi(u_1),\psi(u_2)\rangle}{\|\phi(u_1)\|_2 \cdot \|\psi(u_2)\|_2}.$$

Up to minor details (cf. Appendix G.5), we recover the SimCLR loss with $\phi(u_1) = z_\theta(u_1)$ (no predictor), $\psi(u_2) = z_\theta(u_2)$ (no target network) and $\beta = 1$. We recover the BYOL loss when using a predictor and a target network, *i.e.,* $\phi(u_1) = p_\theta(z_\theta(u_1))$ and $\psi(u_2) = z_\xi(u_2)$ with $\beta = 0$. To evaluate the influence of the target network, the predictor and the coefficient $\beta$, we perform an ablation over them. Results are presented in Table 5b and more details are given in Appendix G.4. The only variant that performs well without negative examples (i.e., with $\beta = 0$) is BYOL, using *both* a bootstrap target network *and* a predictor. Adding the negative pairs to BYOL's loss without re-tuning the temperature parameter hurts its performance. In Appendix G.4, we show that we can add back negative pairs and still match the performance of BYOL with proper tuning of the temperature.

Simply adding a target network to SimCLR already improves performance ($+1.6$ points). This sheds new light on the use of the target network in MoCo [9], where the target network is used to provide more negative examples. Here, we show that by mere stabilization effect, even when using the same number of negative examples, using a target network is beneficial. Finally, we observe that modifying the architecture of $S_\theta$ to include a predictor only mildly affects the performance of SimCLR.

**Relationship with Mean Teacher** Another semi-supervised approach, Mean Teacher (MT) [20], complements a supervised loss on few labels with an additional consistency loss. In [20], this consistency loss is the $\ell_2$ distance between the logits from a *student* network, and those of a temporally averaged version of the student network, called *teacher*. Removing the predictor in BYOL results in an unsupervised version of MT with no classification loss that uses image augmentations instead of the original architectural noise (e.g., dropout). This variant of BYOL collapses (Row 7 of **??**) which suggests that the additional predictor is critical to prevent collapse in an unsupervised scenario.

**Importance of a near-optimal predictor** Table 5b already shows the importance of combining a predictor and a target network: the representation does collapse when either is removed. We further found that we can remove the target network without collapse by making the predictor near-optimal, either by (i) using an optimal *linear* predictor (obtained by linear regression on the current batch) before back-propagating the error through the network ($52.5\%$ top-1 accuracy), or (ii) increasing the learning rate of the predictor ($66.5\%$ top-1). By contrast, increasing the learning rates of both projector *and* predictor (without target network) yields poor results ($\approx 25\%$ top-1). See Appendix J for more details. This seems to indicate that keeping the predictor near-optimal at all times is important to preventing collapse, which may be one of the roles of BYOL's target network.

## 6 Conclusion

We introduced BYOL, a new algorithm for self-supervised learning of image representations. BYOL learns its representation by predicting previous versions of its outputs, without using negative pairs. We show that BYOL achieves state-of-the-art results on various benchmarks. In particular, under the linear evaluation protocol on ImageNet with a ResNet-50 ($1\times$), BYOL achieves a new state of the art and bridges most of the remaining gap between self-supervised methods and the supervised learning baseline of [8]. Using a ResNet-200 ($2\times$), BYOL reaches a top-1 accuracy of $79.6\%$ which improves over the previous state of the art ($76.8\%$) while using $30\%$ fewer parameters.

Nevertheless, BYOL remains dependent on existing sets of augmentations that are specific to vision applications. To generalize BYOL to other modalities (e.g., audio, video, text, . . . ) it is necessary to obtain similarly suitable augmentations for each of them. Designing such augmentations may require significant effort and expertise. Therefore, automating the search for these augmentations would be an important next step to generalize BYOL to other modalities.

## Broader impact

The presented research should be categorized as research in the field of unsupervised learning. This work may inspire new algorithms, theoretical, and experimental investigation. The algorithm presented here can be used for many different vision applications and a particular use may have both positive or negative impacts, which is known as the dual use problem. Besides, as vision datasets could be biased, the representation learned by BYOL could be susceptible to replicate these biases.

## Acknowledgements

The authors would like to thank the following people for their help throughout the process of writing this paper, in alphabetical order: Aaron van den Oord, Andrew Brock, Jason Ramapuram, Jeffrey De Fauw, Karen Simonyan, Katrina McKinney, Nathalie Beauguerlange, Olivier Henaff, Oriol Vinyals, Pauline Luc, Razvan Pascanu, Sander Dieleman, and the DeepMind team. We especially thank Jason Ramapuram and Jeffrey De Fauw, who provided the JAX SimCLR reproduction used throughout the paper.

## Footnotes

[4]Throughout this paper, the term *bootstrap* is used in its idiomatic sense rather than the statistical sense.

[5]While we could directly predict the representation $y$ and not a projection $z$, previous work [8] have empirically shown that using this projection improves performance.

[6]For simplicity we also consider BYOL without normalization (which performs reasonably close to BYOL, see Appendix G.6) nor symmetrization.

[7]The only dependency on batch size in our training pipeline sits within the batch normalization layers.

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
