[Supplementary Material]

## A   Implementation details

**Image augmentations**   BYOL uses the same set of image augmentations as in `SimCLR` [8]. First, a random patch of the image is selected and resized to $224 \times 224$ with a random horizontal flip, followed by a color distortion, consisting of a random sequence of brightness, contrast, saturation, hue adjustments, and an optional grayscale conversion. Finally Gaussian blur and solarization are applied to the patches. Additional details on the image augmentations are in Appendix C.

**Architecture**   We use a convolutional residual network [22] with 50 layers and post-activation (ResNet-50($1\times$) v1) as our base parametric encoders $f_\theta$ and $f_\xi$. We also use deeper (50, 101, 152 and 200 layers) and wider (from $1\times$ to $4\times$) ResNets, as in [67, 48, 8]. Specifically, the representation $y$ corresponds to the output of the final average pooling layer, which has a feature dimension of 2048 (for a width multiplier of $1\times$). As in `SimCLR` [8], the representation $y$ is projected to a smaller space by a *multi-layer perceptron* (MLP) $g_\theta$, and similarly for the target projection $g_\xi$. This MLP consists in a linear layer with output size 4096 followed by batch normalization [68], rectified linear units (ReLU) [69], and a final linear layer with output dimension 256. Contrary to `SimCLR`, the output of this MLP is not batch normalized. The predictor $q_\theta$ uses the same architecture as $g_\theta$.

**Optimization**   We use the `LARS` optimizer [70] with a cosine decay learning rate schedule [71], without restarts, over 1000 epochs, with a warm-up period of 10 epochs. We set the base learning rate to 0.2, scaled linearly [72] with the batch size (LearningRate $= 0.2 \times$ BatchSize/256). In addition, we use a global weight decay parameter of $1.5 \cdot 10^{-6}$ while excluding the biases and batch normalization parameters from both `LARS` adaptation and weight decay. For the target network, the exponential moving average parameter $\tau$ starts from $\tau_{\text{base}} = 0.996$ and is increased to one during training. Specifically, we set $\tau \triangleq 1 - (1 - \tau_{\text{base}}) \cdot (\cos(\pi k/K) + 1)/2$ with $k$ the current training step and $K$ the maximum number of training steps. We use a batch size of 4096 split over 512 Cloud TPU v3 cores. With this setup, training takes approximately 8 hours for a ResNet-50($\times$1). All hyperparameters are summarized in Appendix J; an additional set of hyperparameters for a smaller batch size of 512 is provided in Appendix H.

## B   Algorithm

---
**Algorithm 1:** BYOL: **B**ootstrap **Y**our **O**wn **L**atent

---
**Inputs :**

$\mathcal{D}, \mathcal{T}$, and $\mathcal{T}'$            set of images and distributions of transformations

$\theta, f_\theta, g_\theta$, and $q_\theta$         initial online parameters, encoder, projector, and predictor

$\xi, f_\xi, g_\xi$               initial target parameters, target encoder, and target projector

optimizer              optimizer, updates online parameters using the loss gradient

$K$ and $N$             total number of optimization steps and batch size

$\{\tau_k\}_{k=1}^{K}$ and $\{\eta_k\}_{k=1}^{K}$    target network update schedule and learning rate schedule

1  **for** $k = 1$ **to** $K$ **do**

2      $\mathcal{B} \leftarrow \{x_i \sim \mathcal{D}\}_{i=1}^{N}$             `// sample a batch of` $N$ `images`

3      **for** $x_i \in \mathcal{B}$ **do**

4          $t \sim \mathcal{T}$ and $t' \sim \mathcal{T}'$       `// sample image transformations`

5          $z_1 \leftarrow g_\theta(f_\theta(t(x_i)))$ and $z_2 \leftarrow g_\theta(f_\theta(t'(x_i)))$    `// compute projections`

6          $z_1' \leftarrow g_\xi(f_\xi(t'(x_i)))$ and $z_2' \leftarrow g_\xi(f_\xi(t(x_i)))$    `// compute target projections`

7          $l_i \leftarrow -2 \cdot \left( \frac{\langle q_\theta(z_1), z_1' \rangle}{\|q_\theta(z_1)\|_2 \cdot \|z_1'\|_2} + \frac{\langle q_\theta(z_2), z_2' \rangle}{\|q_\theta(z_2)\|_2 \cdot \|z_2'\|_2} \right)$    `// compute the loss for` $x_i$

8      **end**

9      $\delta\theta \leftarrow \frac{1}{N} \sum_{i=1}^{N} \partial_\theta l_i$         `// compute the total loss gradient w.r.t.` $\theta$

10      $\theta \leftarrow \text{optimizer}(\theta, \delta\theta, \eta_k)$        `// update online parameters`

11      $\xi \leftarrow \tau_k \xi + (1 - \tau_k)\theta$         `// update target parameters`

12 **end**

**Output :** encoder $f_\theta$

---

## C   Image augmentations

During self-supervised training, `BYOL` uses the following image augmentations (which are a subset of the ones presented in [8]):

- random cropping: a random patch of the image is selected, with an area uniformly sampled between 8% and 100% of that of the original image, and an aspect ratio logarithmically sampled between $3/4$ and $4/3$. This patch is then resized to the target size of $224 \times 224$ using bicubic interpolation;

- optional left-right flip;

- color jittering: the brightness, contrast, saturation and hue of the image are shifted by a uniformly random offset applied on all the pixels of the same image. The order in which these shifts are performed is randomly selected for each patch;

- color dropping: an optional conversion to grayscale. When applied, output intensity for a pixel $(r, g, b)$ corresponds to its luma component, computed as $0.2989r + 0.5870g + 0.1140b$;

- Gaussian blurring: for a $224 \times 224$ image, a square Gaussian kernel of size $23 \times 23$ is used, with a standard deviation uniformly sampled over $[0.1, 2.0]$;

- solarization: an optional color transformation $x \mapsto x \cdot \mathbf{1}_{\{x<0.5\}} + (1-x) \cdot \mathbf{1}_{\{x\geq0.5\}}$ for pixels with values in $[0, 1]$.

Augmentations from the sets $\mathcal{T}$ and $\mathcal{T}'$ (introduced in Section 3) are compositions of the above image augmentations in the listed order, each applied with a predetermined probability. The image augmentations parameters are listed in Table 6.

During evaluation, we use a center crop similar to [8]: images are resized to 256 pixels along the shorter side using bicubic resampling, after which a $224 \times 224$ center crop is applied. In both training and evaluation, we normalize color channels by subtracting the average color and dividing by the standard deviation, computed on ImageNet, after applying the augmentations.

| Parameter | $\mathcal{T}$ | $\mathcal{T}'$ |
|---|---|---|
| Random crop probability | 1.0 | 1.0 |
| Flip probability | 0.5 | 0.5 |
| Color jittering probability | 0.8 | 0.8 |
| Brightness adjustment max intensity | 0.4 | 0.4 |
| Contrast adjustment max intensity | 0.4 | 0.4 |
| Saturation adjustment max intensity | 0.2 | 0.2 |
| Hue adjustment max intensity | 0.1 | 0.1 |
| Color dropping probability | 0.2 | 0.2 |
| Gaussian blurring probability | 1.0 | 0.1 |
| Solarization probability | 0.0 | 0.2 |

Table 6: Parameters used to generate image augmentations.

## D   Evaluation on ImageNet training

### D.1   Self-supervised learning evaluation on ImageNet

**Linear evaluation protocol on ImageNet**   As in [48, 74, 8, 37], we use the standard linear evaluation protocol on ImageNet, which consists in training a linear classifier on top of the frozen representation, *i.e.*, without updating the network parameters nor the batch statistics. At training time, we apply spatial augmentations, i.e., random crops with resize to $224 \times 224$ pixels, and random flips. At test time, images are resized to 256 pixels along the shorter side using bicubic resampling, after which a $224 \times 224$ center crop is applied. In both cases, we normalize the color channels by subtracting the average color and dividing by the standard deviation (computed on ImageNet), after applying the augmentations. We optimize the cross-entropy loss using `SGD` with Nesterov momentum over 80 epochs, using a batch size of 1024 and a momentum of 0.9. We do not use any regularization methods such as weight decay, gradient clipping [86], tclip [34], or logits regularization. We finally

sweep over 5 learning rates $\{0.4, 0.3, 0.2, 0.1, 0.05\}$ on a local validation set (10009 images from ImageNet `train` set), and report the accuracy of the best validation hyperparameter on the `test` set (which is the public validation set of the original ILSVRC2012 ImageNet dataset).

**Variant on linear evaluation on ImageNet** In this paragraph only, we deviate from the protocol of [8, 37] and propose another way of performing linear evaluation on top of a frozen representation. This method achieves better performance both in top-1 and top-5 accuracy.

- We replace the spatial augmentations (random crops with resize to $224 \times 224$ pixels and random flips) with the pre-train augmentations of Appendix C. This method was already used in [32] with a different subset of pre-train augmentations.
- We regularize the linear classifier as in [34][8] by clipping the logits using a hyperbolic tangent function

$$\text{tclip}(x) \triangleq \alpha \cdot \tanh(x/\alpha),$$

where $\alpha$ is a positive scalar, and by adding a logit-regularization penalty term in the loss

$$\text{Loss}(x, y) \triangleq \text{cross\_entropy}(\text{tclip}(x), y) + \beta \cdot \text{average}(\text{tclip}(x)^2),$$

where $x$ are the logits, $y$ are the target labels, and $\beta$ is the regularization parameter. We set $\alpha = 20$ and $\beta = 1e{-}2$.

We report in Table 7 the top-1 and top-5 accuracy on ImageNet using this modified protocol. These modifications in the evaluation protocol increase the BYOL's top-1 accuracy from $74.3\%$ to $74.8\%$ with a ResNet-50 ($1\times$).

| Architecture | Pre-train augmentations | Logits regularization | Top-1 | Top-5 |
|---|:---:|:---:|---|---|
| | | | 74.3 | 91.6 |
| ResNet-50 ($1\times$) | ✓ | | 74.4 | 91.8 |
| | | ✓ | 74.7 | 91.8 |
| | ✓ | ✓ | **74.8** | **91.8** |
| | | | 78.6 | 94.2 |
| ResNet-50 ($4\times$) | ✓ | | 78.6 | 94.3 |
| | | ✓ | 78.9 | 94.3 |
| | ✓ | ✓ | **79.0** | **94.5** |
| | | | 79.6 | 94.8 |
| ResNet-200 ($2\times$) | ✓ | | 79.6 | 94.8 |
| | | ✓ | 79.8 | 95.0 |
| | ✓ | ✓ | **80.0** | **95.0** |

Table 7: Different linear evaluation protocols on ResNet architectures by either replacing the spatial augmentations with pre-train augmentations, or regularizing the linear classifier. No pre-train augmentations and no logits regularization correspond to the evaluation protocol of the main paper, which is the same as in [8, 37].

**Semi-supervised learning on ImageNet** We follow the semi-supervised learning protocol of [8, 77]. We first initialize the network with the parameters of the pretrained representation, and fine-tune it with a subset of ImageNet labels. At training time, we apply spatial augmentations, i.e., random crops with resize to $224 \times 224$ pixels and random flips. At test time, images are resized to 256 pixels along the shorter side using bicubic resampling, after which a $224 \times 224$ center crop is applied. In both cases, we normalize the color channels by subtracting the average color and dividing by the standard deviation (computed on ImageNet), after applying the augmentations. We optimize the cross-entropy loss using `SGD` with Nesterov momentum. We used a batch size of 1024, a momentum of 0.9. We do not use any regularization methods such as weight decay, gradient clipping [86], tclip [34], or logits rescaling. We sweep over the learning rate $\{0.01, 0.02, 0.05, 0.1, 0.005\}$ and the number of epochs $\{30, 50\}$ and select the hyperparameters achieving the best performance on our local validation set to report test performance.

(a) Top-1 accuracy

(b) Top-5 accuracy

Figure 4: Semi-supervised training with a fraction of ImageNet labels on a ResNet-50 ($\times$1).

| *Supervised:* | | | *Semi-supervised* (100%): | | |
|---|---|---|---|---|---|
| Method | Top-1 | Top-5 | Method | Top-1 | Top-5 |
| Supervised[8] | 76.5 | – | SimCLR [8] | 76.0 | 93.1 |
| AutoAugment [87] | 77.6 | 93.8 | SimCLR (repro) | 76.5 | 93.5 |
| MaxUp [75] | **78.9** | **94.2** | BYOL | **77.7** | **93.9** |

Table 8: Semi-supervised training with the full ImageNet on a ResNet-50 ($\times$1). We also report other fully supervised methods for extensive comparisons.

In Table 2 presented in the main text, we fine-tune the representation over the 1% and 10% ImageNet splits from [8] with various ResNet architectures.

In Figure 4, we fine-tune the representation over 1%, 2%, 5%, 10%, 20%, 50%, and 100% of the ImageNet dataset as in [32] with a ResNet-50 (1$\times$) architecture, and compare them with a supervised baseline and a fine-tuned SimCLR representation. In this case and contrary to Table 2 we don't reuse the splits from SimCLR but we create our own via a balanced selection. In this setting, we observed that tuning a BYOL representation always outperforms a supervised baseline trained from scratch. In Figure 5, we then fine-tune the representation over multiple ResNet architectures. We observe that the largest networks are prone to overfitting as they are outperformed by ResNets with identical depth but smaller scaling factor. This overfitting is further confirmed when looking at the training and evaluation loss: large networks have lower training losses, but higher validation losses than some of their slimmer counterparts. Regularization methods are thus recommended when tuning on large architectures.

Finally, we fine-tune the representation over the full ImageNet dataset. We report the results in Table 8 along with supervised baselines trained on ImageNet. We observe that fine-tuning the SimCLR checkpoint does not yield better results (in our reproduction, which matches the results reported in the original paper [8]) than using a random initialization (76.5 top-1). Instead, BYOL's initialization checkpoint leads to a high final score (77.7 top-1), higher than the vanilla supervised baseline of [8], matching the strong supervised baseline of AutoAugment[87] but still 1.2 points below the stronger supervised baseline [75], which uses advanced supervised learning techniques.

## D.2 Linear evaluation on larger architectures and supervised baselines

Here we investigate the performance of BYOL with deeper and wider ResNet architectures. We compare ourselves to the best supervised baselines from [8] when available (rightmost column in table 9), which are also presented in Figure 1. Importantly, we close in on those baselines using

Figure 5: Semi-supervised training with a fraction of ImageNet labels on multiple ResNets architecture pretrained with BYOL. Note that large networks are facing overfitting problems.

| Architecture | Multiplier | Weights | BYOL | | Supervised (ours) | | Supervised [8] |
| | | | Top-1 | Top-5 | Top-1 | Top-5 | Top-1 |
|---|---|---|---|---|---|---|---|
| ResNet-50 | $1\times$ | 24M | 74.3 | 91.6 | 76.4 | 92.9 | 76.5 |
| ResNet-101 | $1\times$ | 43M | 76.4 | 93.0 | 78.0 | 94.0 | - |
| ResNet-152 | $1\times$ | 58M | 77.3 | 93.7 | 79.1 | 94.5 | - |
| ResNet-200 | $1\times$ | 63M | 77.8 | 93.9 | 79.3 | 94.6 | - |
| ResNet-50 | $2\times$ | 94M | 77.4 | 93.6 | 79.9 | 95.0 | 77.8 |
| ResNet-101 | $2\times$ | 170M | 78.7 | 94.3 | 80.3 | 95.0 | - |
| ResNet-50 | $3\times$ | 211M | 78.2 | 93.9 | 80.2 | 95.0 | - |
| ResNet-152 | $2\times$ | 232M | 79.0 | 94.6 | 80.6 | 95.3 | - |
| ResNet-200 | $2\times$ | 250M | **79.6** | **94.9** | 80.1 | 95.2 | - |
| ResNet-50 | $4\times$ | 375M | 78.6 | 94.2 | 80.7 | **95.3** | 78.9 |
| ResNet-101 | $3\times$ | 382M | 78.4 | 94.2 | 80.7 | **95.3** | - |
| ResNet-152 | $3\times$ | 522M | 79.5 | 94.6 | **80.9** | 95.2 | - |

Table 9: Linear evaluation of BYOL on ImageNet using larger encoders.
Top-1 and top-5 accuracies are reported in %.

the ResNet-50 ($2\times$) and the ResNet-50 ($4\times$) architectures, where we are within $0.4$ accuracy points of the supervised performance. To the best of our knowledge, this is the first time that the gap to supervised has been closed to such an extent using a self-supervised method under the linear evaluation protocol. Therefore, in order to ensure fair comparison, and suspecting that the supervised baselines' performance in [8] could be even further improved with appropriate data augmentations, we also report on our own reproduction of strong supervised baselines. We use RandAugment [87] data augmentation for all large ResNet architectures (which are all version 1, as per [22]). We train our supervised baselines for up to 200 epochs, using SGD with a Nesterov momentum value of $0.9$, a cosine-annealed learning rate after a $5$ epochs linear warmup period, weight decay with a value of $1e - 4$, and a label smoothing [88] value of $0.1$. Results are presented in Figure 6.

Figure 6: Results for linear evaluation of BYOL compared to fully supervised baselines with various ResNet architectures. Our supervised baselines are ran with RandAugment [87] augmentations.

# E Transfer to other datasets

## E.1 Datasets

| Dataset | Classes | Original train examples | Train examples | Valid. examples | Test examples | Accuracy measure | Test provided |
|---|---|---|---|---|---|---|---|
| ImageNet [21] | 1000 | 1281167 | 1271158 | 10009 | 50000 | Top-1 accuracy | - |
| Food101 [89] | 101 | 75750 | 68175 | 7575 | 25250 | Top-1 accuracy | - |
| CIFAR-10 [78] | 10 | 50000 | 45000 | 5000 | 10000 | Top-1 accuracy | - |
| CIFAR-100 [78] | 100 | 50000 | 44933 | 5067 | 10000 | Top-1 accuracy | - |
| Birdsnap [90] | 500 | 47386 | 42405 | 4981 | 2443 | Top-1 accuracy | - |
| Sun397 (split 1) [79] | 397 | 19850 | 15880 | 3970 | 19850 | Top-1 accuracy | - |
| Cars [91] | 196 | 8144 | 6494 | 1650 | 8041 | Top-1 accuracy | - |
| Aircraft [92] | 100 | 3334 | 3334 | 3333 | 3333 | Mean per-class accuracy | Yes |
| PASCAL-VOC2007 [80] | 20 | 5011 | 2501 | 2510 | 4952 | 11-point mAP / AP50 | - |
| PASCAL-VOC2012 [80] | 21 | 10582 | – | 2119 | 1449 | Mean IoU | - |
| DTD (split 1) [81] | 47 | 1880 | 1880 | 1880 | 1880 | Top-1 accuracy | Yes |
| Pets [93] | 37 | 3680 | 2940 | 740 | 3669 | Mean per-class accuracy | - |
| Caltech-101 [94] | 101 | 3060 | 2550 | 510 | 6084 | Mean per-class accuracy | - |
| Places365 [73] | 365 | 1803460 | 1803460 | – | 36500 | Top-1 accuracy | - |
| Flowers [95] | 102 | 1020 | 1020 | 1020 | 6149 | Mean per-class accuracy | Yes |

Table 10: Characteristics of image datasets used in transfer learning. When an official test split with labels is not publicly available, we use the official validation split as test set, and create a held-out validation set from the training examples.

We perform transfer via linear classification and fine-tuning on the same set of datasets as in [8], namely Food-101 dataset [89], CIFAR-10 [78] and CIFAR-100 [78], Birdsnap [90], the SUN397 scene dataset [79], Stanford Cars [91], FGVC Aircraft [92], the PASCAL VOC 2007 classification task [80], the Describable Textures Dataset (DTD) [81], Oxford-IIIT Pets [93], Caltech-101 [94], and Oxford 102 Flowers [95]. As in [8], we used the validation sets specified by the dataset creators to select hyperparameters for FGVC Aircraft, PASCAL VOC 2007, DTD, and Oxford 102 Flowers. On other datasets, we use the validation examples as test set, and hold out a subset of the training examples that we use as validation set. We use standard metrics for each datasets:

- *Top-1*: We compute the proportion of correctly classified examples.

- *Mean per class*: We compute the top-1 accuracy for each class separately and then compute the empirical mean over the classes.

- *Point 11-mAP*: We compute the empirical mean *average precision* as defined in [80].

- *Mean IoU:* We compute the empirical mean Intersection-Over-Union as defined in [80].

- *AP50:* We compute the Average Precision as defined in [80].

We detail the validation procedures for some specific datasets:

- For Sun397 [79], the original dataset specifies 10 train/test splits, all of which contain 50 examples/images of 397 different classes. We use the first train/test split. The original dataset specifies no validation split and therefore, the training images have been further subdivided into 40 images per class for the train split and 10 images per class for the valid split.

- For Birdsnap [90], we use a random selection of valid images with the same number of images per category as the test split.

- For DTD [81], the original dataset specifies 10 train/validation/test splits, we only use the first split.

- For Caltech-101 [94], the original does not dataset specifies any train/test splits. We have followed the approach used in [96]: This file defines datasets for 5 random splits of 25 training images per category, with 5 validation images per category and the remaining images used for testing.

- For ImageNet, we took the last 10009 last images of the official tensorflow ImageNet split.

- For Oxford-IIIT Pets, the valid set consists of 20 randomly selected images per class.

Information about the dataset are summarized in Table 10.

## E.2 Transfer via linear classification

We follow the linear evaluation protocol of [48, 74, 8] that we detail next for completeness. We train a regularized multinomial logistic regression classifier on top of the frozen representation, i.e., with frozen pretrained parameters and without re-computing batch-normalization statistics. In training and testing, we do not perform any image augmentations; images are resized to 224 pixels along the shorter side using bicubic resampling and then normalized with ImageNet statistics. Finally, we minimize the cross-entropy objective using LBFGS with $\ell_2$-regularization, where we select the regularization parameters from a range of 45 logarithmically-spaced values between $10^{-6}$ and $10^5$. After choosing the best-performing hyperparameters on the validation set, the model is retrained on combined training and validation images together, using the chosen parameters. The final accuracy is reported on the test set.

## E.3 Transfer via fine-tuning

We follow the same fine-tuning protocol as in [32, 48, 76, 8] that we also detail for completeness. Specifically, we initialize the network with the parameters of the pretrained representation. At training time, we apply spatial transformation, i.e., random crops with resize to $224 \times 224$ pixels and random flips. At test time, images are resized to 256 pixels along the shorter side using bicubic resampling, after which a $224 \times 224$ center crop is extracted. In both cases, we normalize the color channels by subtracting the average color and dividing by the standard deviation (computed on ImageNet), after applying the augmentations. We optimize the loss using SGD with Nesterov momentum for 20000 steps with a batch size of 256 and with a momentum of 0.9. We set the momentum parameter for the batch normalization statistics to $\max(1 - 10/s, 0.9)$ where $s$ is the number of steps per epoch. The learning rate and weight decay are selected respectively with a grid of seven logarithmically spaced learning rates between 0.0001 and 0.1, and 7 logarithmically-spaced values of weight decay between $10^{-6}$ and $10^{-3}$, as well as no weight decay. These values of weight decay are divided by the learning rate. After choosing the best-performing hyperparameters on the validation set, the model is retrained on combined training and validation images together, using the chosen parameters. The final accuracy is reported on the test set.

## E.4 Implementation details for semantic segmentation

We use the same fully-convolutional network (FCN)-based [7] architecture as [9]. The backbone consists of the convolutional layers in ResNet-50. The $3 \times 3$ convolutions in the conv5 blocks use dilation 2 and stride 1. This is followed by two extra $3 \times 3$ convolutions with 256 channels, each followed by batch normalization and ReLU activations, and a $1 \times 1$ convolution for per-pixel classification. The dilation is set to 6 in the two extra $3 \times 3$ convolutions. The total stride is 16 (FCN-16s [7]).

We train on the `train_aug2012` set and report results on `val2012`. Hyperparameters are selected on a 2119 images held-out validation set. We use a standard per-pixel softmax cross-entropy loss to train the FCN. Training is done with random scaling (by a ratio in $[0.5, 2.0]$), cropping, and horizontal flipping. The crop size is 513. Inference is performed on the $[513, 513]$ central crop. For training we use a batch size of 16 and weight decay of 0.0001. We select the base learning rate by sweeping across 5 logarithmically spaced values between $10^{-3}$ and $10^{-1}$. The learning rate is multiplied by 0.1 at the 70-th and 90-th percentile of training. We train for 30000 iterations, and average the results on 5 seeds.

## E.5 Implementation details for object detection

For object detection, we follow prior work on Pascal detection transfer [40, 23] wherever possible. We use a Faster R-CNN [82] detector with a R50-C4 backbone with a frozen representation. The R50-C4 backbone ends with the conv4 stage of a ResNet-50, and the box prediction head consists of the conv5 stage (including global pooling). We preprocess the images by applying multi-scale augmentation (rescaling the image so its longest edge is between 480 and 1024 pixels) but no other augmentation. We use an asynchronous SGD optimizer with 9 workers and train for 1.5M steps. We used an initial learning rate of $10^{-3}$, which is reduced to $10^{-4}$ at 1M steps and to $10^{-5}$ at 1.2M steps.

### E.6 Implementation details for depth estimation

For depth estimation, we follow the same protocol as in [83], and report its core components for completeness. We use a standard ResNet-50 backbone and feed the conv5 features into 4 fast up-projection blocks with respective filter sizes 512, 256, 128, and 64. We use a reverse Huber loss function for training [83, 97].

The original NYU Depth v2 frames of size $[640, 480]$ are down-sampled by a factor 0.5 and center-cropped to $[304, 228]$ pixels. Input images are randomly horizontally flipped and the following color transformations are applied:

- *Grayscale* with an application probability of 0.3.
- *Brightness* with a maximum brightness difference of 0.1255.
- *Saturation* with a saturation factor randomly picked in the interval $[0.5, 1.5]$.
- *Hue* with a hue adjustment factor randomly picked in the interval $[-0.2, 0.2]$.

We train for 7500 steps with batch size 256, weight decay 0.0005, and learning rate 0.16 (scaled linearly from the setup of [83] to account for the bigger batch size).

### E.7 Further comparisons on PASCAL and NYU v2 Depth

For completeness, Table 11 and 12 extends Table 4 with other published baselines which use comparable networks. We see that in almost all settings, BYOL outperforms these baselines, even when those baselines use more data or deeper models. One notable exception is RMS error for NYU Depth prediction, which is a metric that's sensitive to outliers. The reason for this is unclear, but one possibility is that the network is producing higher-variance predictions due to being more confident about a test-set scene's similarities with those in the training set.

| Method | $AP_{50}$ | mIoU |
|---|---|---|
| Supervised-IN [9] | 74.4 | 74.4 |
| RelPos [23], by [40][*] | 66.8 | - |
| Multi-task [40][*] | 70.5 | - |
| LocalAgg [98] | 69.1 | - |
| MoCo [9] | 74.9 | 72.5 |
| MoCo + IG-1B [9] | 75.6 | 73.6 |
| CPC[32][**] | 76.6 | - |
| SimCLR (repro) | 75.2 | 75.2 |
| BYOL (ours) | **77.5** | **76.3** |

Table 11: Transfer results in semantic segmentation and object detection.

[*] uses a larger model (ResNet-101). [**] uses an even larger model (ResNet-161).

| Method | Higher better | | | Lower better | |
|---|---|---|---|---|---|
| | pct.$<1.25$ | pct.$<1.25^2$ | pct.$<1.25^3$ | rms | rel |
| Supervised-IN [83] | 81.1 | 95.3 | 98.8 | 0.573 | **0.127** |
| RelPos [23], by [40][*] | 80.6 | 94.7 | 98.3 | **0.399** | 0.146 |
| Color [41], by [40][*] | 76.8 | 93.5 | 97.7 | 0.444 | 0.164 |
| Exemplar [46, 40][*] | 71.3 | 90.6 | 96.5 | 0.513 | 0.191 |
| Mot. Seg. [99], by [40][*] | 74.2 | 92.4 | 97.4 | 0.473 | 0.177 |
| Multi-task [40][*] | 79.3 | 94.2 | 98.1 | 0.422 | 0.152 |
| SimCLR (repro) | 83.3 | 96.5 | 99.1 | 0.557 | 0.134 |
| BYOL (ours) | **84.6** | **96.7** | **99.1** | 0.541 | 0.129 |

Table 12: Transfer results on NYU v2 depth estimation.

# F    Pretraining on Places 365

To ascertain that `BYOL` learns good representations on other datasets, we applied our representation learning protocol on the scene recognition dataset Places365-Standard [73] before performing linear evaluation. This dataset contains 1.80 million training images and 36500 validation images with labels, making it roughly similar to ImageNet in scale. We reuse the *exact* same parameters as in Section 4 and train the representation for 1000 epochs, using `BYOL` and our `SimCLR` reproduction. Results for the linear evaluation setup (using the protocol of Appendix D.1 for ImageNet and Places365, and that of Appendix E on other datasets) are reported in Table 13.

Interestingly, the representation trained by using `BYOL` on Places365 (BYOL-PL) consistently outperforms that of `SimCLR` on the same dataset, but underperforms the `BYOL` representation trained on ImageNet (BYOL-IN) on all tasks except Places365 and SUN397 [79], another scene understanding dataset. Interestingly, all three unsupervised representation learning methods achieve a relatively high performance on the Places365 task; for comparison, reference [73] (in its linked repository) reports a top-1 accuracy of $55.2\%$ for a ResNet-50v2 trained from scratch using labels on this dataset.

| Method | Places365 | ImageNet | Food101 | CIFAR10 | CIFAR100 | Birdsnap | SUN397 | Cars | Aircraft | DTD | Pets | Caltech-101 | Flowers |
|---|---|---|---|---|---|---|---|---|---|---|---|---|---|
| BYOL-IN | 51.0 | 74.3 | 75.3 | 91.3 | 78.4 | 57.3 | 62.6 | 67.2 | 60.6 | 76.5 | 90.4 | 94.3 | 96.1 |
| BYOL-PL | 53.2 | 58.5 | 64.7 | 84.5 | 66.1 | 28.8 | 64.2 | 55.6 | 55.9 | 68.5 | 66.1 | 84.3 | 90.0 |
| SimCLR-PL | 53.0 | 56.5 | 61.7 | 80.8 | 61.1 | 21.2 | 62.5 | 40.1 | 44.3 | 64.3 | 59.4 | 77.1 | 85.9 |

Table 13: Transfer learning results (linear evaluation, ResNet-50) from Places365 (PL). For comparison purposes, we also report the results from `BYOL` trained on ImageNet (BYOL-IN).

# G    Additional ablation results

To extend on the above results, we explore how other network parameters may impact `BYOL`'s performance. We iterate over multiple weight decays, learning rates, and projector/encoder architectures to observe that small hyperparameter changes do not drastically alter the final score. We note that removing the weight decay in either `BYOL` or `SimCLR` leads to network divergence, emphasizing the need for weight regularization in the self-supervised setting. Furthermore, we observe that changing the scaling factor in the network initialization [85] did not impact the performance (higher than $72\%$ top-1 accuracy).

We use the same experimental setup as in Section 5, *i.e.,* 300 epochs, averaged over 3 seeds with the initial learning rate set to 0.3, the batch size to 4096, the weight decay to $10^{-6}$ and the base target decay rate $\tau_{\text{base}}$ to 0.99 unless specified otherwise. Confidence intervals correspond to the half-difference between the maximum and minimum score of these seeds; we omit them for half-differences lower than 0.25 accuracy points.

## G.1    Architecture settings

Table 14 shows the influence of projector and predictor architecture on `BYOL`. We examine the effect of different depths for both the projector and predictor, as well as the effect of the projection size. We do not apply a ReLU activation nor a batch normalization on the final linear layer of our MLPs such that a depth of 1 corresponds to a linear layer. Using the default projector and predictor of depth 2 yields the best performance.

Table 15a shows the influence of the initial learning rate on `BYOL`. Note that the optimal value depends on the number of training epochs. Table 15b displays the influence of the weight decay on `BYOL`.

## G.2    Batch size

We run a sweep over the batch size for both `BYOL` and our reproduction of `SimCLR`. As explained in Section 5, when reducing the batch size by a factor $N$, we average gradients over $N$ consecutive steps and update the target network once every $N$ steps. We report in Table 16, the performance of both our reproduction of `SimCLR` and `BYOL` for batch sizes between 4096 (BYOL and SimCLR default) down to 64. We observe that the performance of `SimCLR` deteriorates faster than the one of `BYOL`

| Proj. $g_\theta$ depth | Pred. $q_\theta$ depth | Top-1 | Top-5 |
|---|---|---|---|
| | 1 | 61.9 | 86.0 |
| 1 | 2 | 65.0 | 86.8 |
| | 3 | 65.7 | 86.8 |
| | 1 | 71.5 | 90.7 |
| 2 | 2 | **72.5** | **90.8** |
| | 3 | 71.4 | 90.4 |
| | 1 | 71.4 | 90.4 |
| 3 | 2 | 72.1 | 90.5 |
| | 3 | 72.1 | 90.5 |

(a) Projector and predictor depth (i.e. the number of Linear layers).

| Projector $g_\theta$ output dim | Top-1 | Top-5 |
|---|---|---|
| 16 | $69.9_{\pm0.3}$ | 89.9 |
| 32 | 71.3 | 90.6 |
| 64 | 72.2 | 90.9 |
| 128 | 72.5 | 91.0 |
| 256 | 72.5 | 90.8 |
| 512 | **72.6** | **91.0** |

(b) Projection dimension.

Table 14: Effect of architectural settings where top-1 and top-5 accuracies are reported in %.

| Learning rate | Top-1 | Top-5 |
|---|---|---|
| 0.01 | $34.8_{\pm3.0}$ | $60.8_{\pm3.2}$ |
| 0.1 | 65.0 | 87.0 |
| 0.2 | 71.7 | 90.6 |
| 0.3 | **72.5** | **90.8** |
| 0.4 | 72.3 | 90.6 |
| 0.5 | 71.5 | 90.1 |
| 1 | 69.4 | 89.2 |

(a) Base learning rate.

| Weight decay coefficient | Top-1 | Top-5 |
|---|---|---|
| $1 \cdot 10^{-7}$ | 72.1 | 90.4 |
| $5 \cdot 10^{-7}$ | **72.6** | **91.0** |
| $1 \cdot 10^{-6}$ | 72.5 | 90.8 |
| $5 \cdot 10^{-6}$ | $71.0_{\pm0.3}$ | 90.0 |
| $1 \cdot 10^{-5}$ | $69.6_{\pm0.4}$ | 89.3 |

(b) Weight decay.

Table 15: Effect of learning rate and weight decay. We note that `BYOL`'s performance is quite robust within a range of hyperparameters. We also observe that setting the weight decay to zero may lead to unstable results (as in `SimCLR`).

which stays mostly constant for batch sizes larger than 256. We believe that the performance at batch size 256 could match the performance of the large 4096 batch size with proper parameter tuning when accumulating the gradient. We think that the drop in performance at batch size 64 in table 16 is mainly related to the ill behaviour of batch normalization at low batch sizes [100].

| Batch size | Top-1 | | Top-5 | |
|---|---|---|---|---|
| | BYOL (ours) | SimCLR (repro) | BYOL (ours) | SimCLR (repro) |
| 4096 | **72.5** | 67.9 | **90.8** | 88.5 |
| 2048 | 72.4 | 67.8 | 90.7 | 88.5 |
| 1024 | 72.2 | 67.4 | 90.7 | 88.1 |
| 512 | 72.2 | 66.5 | 90.8 | 87.6 |
| 256 | 71.8 | $64.3_{\pm2.1}$ | 90.7 | $86.3_{\pm1.0}$ |
| 128 | $69.6_{\pm0.5}$ | 63.6 | 89.6 | 85.9 |
| 64 | $59.7_{\pm1.5}$ | $59.2_{\pm2.9}$ | $83.2_{\pm1.2}$ | $83.0_{\pm1.9}$ |

Table 16: Influence of the batch size.

## G.3 Image augmentations

Table 17 compares the impact of individual image transformations on `BYOL` and `SimCLR`. `BYOL` is more resilient to changes of image augmentations across the board. For completeness, we also include an ablation with symmetric parameters across both views; for this ablation, we use a Gaussian blurring w.p. of 0.5 and a solarization w.p. of 0.2 for both $\mathcal{T}$ and $\mathcal{T}'$, and recover very similar results compared to our baseline choice of parameters.

| Image augmentation | Top-1 | | Top-5 | |
|---|---|---|---|---|
| | BYOL (ours) | SimCLR (repro) | BYOL (ours) | SimCLR (repro) |
| Baseline | **72.5** | 67.9 | **90.8** | 88.5 |
| Remove flip | 71.9 | 67.3 | 90.6 | 88.2 |
| Remove blur | 71.2 | 65.2 | 90.3 | 86.6 |
| Remove color (jittering and grayscale) | $63.4_{\pm 0.7}$ | 45.7 | $85.3_{\pm 0.5}$ | 70.6 |
| Remove color jittering | 71.8 | 63.7 | 90.7 | 85.9 |
| Remove grayscale | 70.3 | 61.9 | 89.8 | 84.1 |
| Remove blur in $\mathcal{T}'$ | 72.4 | 67.5 | 90.8 | 88.4 |
| Remove solarize in $\mathcal{T}'$ | 72.3 | 67.7 | 90.8 | 88.2 |
| Remove blur and solarize in $\mathcal{T}'$ | 72.2 | 67.4 | 90.8 | 88.1 |
| Symmetric blurring/solarization | 72.5 | 68.1 | 90.8 | 88.4 |
| Crop only | $59.4_{\pm 0.3}$ | $40.3_{\pm 0.3}$ | 82.4 | $64.8_{\pm 0.4}$ |
| Crop and flip only | $60.1_{\pm 0.3}$ | 40.2 | $83.0_{\pm 0.3}$ | 64.8 |
| Crop and color only | 70.7 | 64.2 | 90.0 | 86.2 |
| Crop and blur only | $61.1_{\pm 0.3}$ | 41.7 | 83.9 | 66.4 |

Table 17: Ablation on image transformations.

| Loss weight $\beta$ | Temperature $\alpha$ | Top-1 | Top-5 |
|---|---|---|---|
| 0 | 0.1 | 72.5 | 90.8 |
| | 0.01 | 72.2 | 90.7 |
| | 0.1 | 72.4 | 90.9 |
| | 0.3 | **72.7** | 91.0 |
| 0.1 | 1 | 72.6 | 90.9 |
| | 3 | 72.5 | 90.9 |
| | 10 | 72.5 | 90.9 |
| | 0.01 | 70.9 | 90.2 |
| | 0.1 | 72.0 | 90.8 |
| | 0.3 | **72.7** | **91.2** |
| 0.5 | 1 | 72.7 | 91.1 |
| | 3 | 72.6 | 91.1 |
| | 10 | 72.5 | 91.0 |
| | 0.01 | $53.9_{\pm 0.5}$ | $77.5_{\pm 0.5}$ |
| | 0.1 | 70.9 | 90.3 |
| | 0.3 | **72.7** | 91.1 |
| 1 | 1 | **72.7** | 91.1 |
| | 3 | 72.6 | 91.0 |
| | 10 | 72.6 | 91.1 |

Table 18: Top-1 accuracy in % under linear evaluation protocol at 300 epochs of sweep over the temperature $\alpha$ and the dispersion term weight $\beta$ when using a predictor and a target network.

## G.4 Details on the relation to contrastive methods

As mentioned in Section 5, the BYOL loss Eq. 2 can be derived from the InfoNCE loss

$$\text{InfoNCE}_{\theta}^{\alpha,\beta} \triangleq \frac{2}{B}\sum_{i=1}^{B} S_{\theta}(v_i, v_i') - \frac{2\alpha \cdot \beta}{B}\sum_{i=1}^{B} \ln\left(\sum_{j\neq i}\exp\frac{S_{\theta}(v_i, v_j)}{\alpha} + \sum_{j}\exp\frac{S_{\theta}(v_i, v_j')}{\alpha}\right), \quad (5)$$

with

$$S_{\theta}(u_1, u_2) \triangleq \frac{\langle \phi(u_1), \psi(u_2)\rangle}{\|\phi(u_1)\|_2 \cdot \|\psi(u_2)\|_2}. \quad (6)$$

The InfoNCE loss, introduced in [10], can be found in factored form in [84] as

$$\text{InfoNCE}_\theta \triangleq \frac{1}{B} \sum_{i=1}^{B} \ln \frac{f(v_i, v_i')}{\frac{1}{B} \sum_j \exp f(v_i, v_j')}. \tag{7}$$

As in `SimCLR` [8] we also use negative examples given by $(v_i, v_j)_{j \neq i}$ to get

$$\frac{1}{B} \sum_{i=1}^{B} \ln \frac{\exp f(v_i, v_i')}{\frac{1}{B} \sum_{j \neq i} \exp f(v_i, v_j) + \frac{1}{B} \sum_j \exp f(v_i, v_j')} \tag{8}$$

$$= \ln B + \frac{1}{B} \sum_{i=1}^{B} f(v_i, v_i') - \frac{1}{B} \sum_{i=1}^{B} \ln \left( \sum_{j \neq i} \exp f(v_i, v_j) + \sum_j \exp f(v_i, v_j') \right). \tag{9}$$

To obtain Eq. 5 from Eq. 9, we subtract $\ln B$ (which is independent of $\theta$), multiply by $2\alpha$, take $f(x,y) = S_\theta(x,y)/\alpha$ and finally multiply the second (negative examples) term by $\beta$. Using $\beta = 1$ and dividing by $2\alpha$ gets us back to the usual InfoNCE loss as used by `SimCLR`.

In our ablation in Table 5b, we set the temperature $\alpha$ to its best value in the `SimCLR` setting (i.e., $\alpha = 0.1$). With this value, setting $\beta$ to 1 (which adds negative examples), in the BYOL setting (i.e., with both a predictor and a target network) hurts the performances. In Table 18, we report results of a sweep over both the temperature $\alpha$ and the weight parameter $\beta$ with a predictor and a target network where BYOL corresponds to $\beta = 0$. No run significantly outperforms BYOL and some values of $\alpha$ and $\beta$ hurt the performance. While the best temperature for `SimCLR` (without the target network and a predictor) is $0.1$, after adding a predictor and a target network the best temperature $\alpha$ is higher than $0.3$.

Using a target network in the loss has two effects: stopping the gradient through the prediction targets and stabilizing the targets with averaging. Stopping the gradient through the target change the objective while averaging makes the target stable and stale. In Table 5b we only shows results of the ablation when either using the online network as the prediction target (and flowing the gradient through it) or with a target network (both stopping the gradient into the prediction targets and computing the prediction targets with a moving average of the online network). We shown in Table 5b that using a target network is beneficial but it has two distinct effects we would like to understand from which effect the improvement comes from. We report in Table 19 the results already in Table 5b but also when the prediction target is computed with a stop gradient of the online network (the gradient does not flow into the prediction targets). This shows that making the prediction targets stable and stale is the main cause of the improvement rather than the change in the objective due to the stop gradient.

### G.5 `SimCLR` baseline of Section 5

The `SimCLR` baseline in Section 5 ($\beta = 1$, without predictor nor target network) is slightly different from the original one in [8]. First we multiply the original loss by $2\alpha$. For comparaison here is the original `SimCLR` loss,

$$\text{InfoNCE}_\theta \triangleq \frac{1}{B} \sum_{i=1}^{B} \frac{S_\theta(v_i, v_i')}{\alpha} - \frac{1}{B} \sum_{i=1}^{B} \ln \left( \sum_{j \neq i} \exp \frac{S_\theta(v_i, v_j)}{\alpha} + \sum_j \exp \frac{S_\theta(v_i, v_j')}{\alpha} \right). \tag{10}$$

Note that this multiplication by $2\alpha$ matters as the `LARS` optimizer is not completely invariant with respect to the scale of the loss. Indeed, `LARS` applies a preconditioning to gradient updates on all weights, except for biases and batch normalization parameters. Updates on preconditioned weights are invariant by multiplicative scaling of the loss. However, the bias and batch normalization parameter updates remain sensitive to multiplicative scaling of the loss.

We also increase the original `SimCLR` hidden and output size of the projector to respectively 4096 and 256. In our reproduction of `SimCLR`, these three combined changes improves the top-1 accuracy at 300 epochs from $67.9\%$ (without the changes) to $69.2\%$ (with the changes).

| Method | Predictor | Target parameters | $\beta$ | Top-1 |
|---|---|---|---|---|
| BYOL | ✓ | $\xi$ | 0 | **72.5** |
| | ✓ | $\xi$ | 1 | 70.9 |
| | | $\xi$ | 1 | 70.7 |
| | ✓ | $sg(\theta)$ | 1 | 70.2 |
| SimCLR | | $\theta$ | 1 | 69.4 |
| | ✓ | $sg(\theta)$ | 1 | 70.1 |
| | | $sg(\theta)$ | 1 | 69.2 |
| | ✓ | $\theta$ | 1 | 69.0 |
| | ✓ | $sg(\theta)$ | 0 | 5.5 |
| | ✓ | $\theta$ | 0 | 0.3 |
| | | $\xi$ | 0 | 0.2 |
| | | $sg(\theta)$ | 0 | 0.1 |
| | | $\theta$ | 0 | 0.1 |

Table 19: Top-1 accuracy in %, under linear evaluation protocol at 300 epochs, of intermediate variants between BYOL and SimCLR (with caveats discussed in Appendix G.5). sg means stop gradient.

(a) Representation $\ell_2$-norm

(b) Projection $\ell_2$-norm

Figure 7: Effect of normalization on the $\ell_2$ norm of network outputs.

| Normalization | Top-1 | Top-5 |
|---|---|---|
| $\ell_2$-norm | **72.5** | **90.8** |
| LAYERNORM | $72.5_{\pm 0.4}$ | 90.1 |
| No normalization | 67.4 | 87.1 |
| BATCHNORM | 65.3 | 85.3 |

Table 20: Top-1 accuracy in % under linear evaluation protocol at 300 epochs for different normalizations in the loss.

### G.6 Ablation on the normalization in the loss function

BYOL minimizes a squared error between the $\ell_2$-normalized prediction and target. We report results of BYOL at 300 epochs using different normalization function and no normalization at all. More precisely, given batch of prediction and targets in $\mathbb{R}^d$, $(p_i, t_i)_{i \leq B}$ with $B$ the batch size, BYOL uses the loss function $\frac{1}{B} \sum_{i=1}^{B} \|n_{\ell_2}(p_i) - n_{\ell_2}(z_i)\|_2^2$ with $n_{\ell_2} : x \to x/\|x\|_2$. We run BYOL with other normalization functions: non-trainable batch-normalization and layer-normalization and no normalization. We divide the batch normalization and layer normalization by $\sqrt{d}$ to have a consistent scale with the $\ell_2$-normalization. We report results in Table 20 where $\ell_2$, LAYERNORM, no normalization and BATCHNORM respectively denote using $n_{\ell_2}$, $n_{\text{BN}}$, $n_{\text{LN}}$ and $n_{\text{ID}}$ with

$$n_{\text{BN}\,i}^j : x \to \frac{x_i^j - \mu_{\text{BN}}^j(x)}{\sigma_{\text{BN}}^j(x) \cdot \sqrt{d}} \, , \quad n_{\text{LN}\,i}^j : x \to \frac{x_i^j - \mu_{\text{LN}\,i}(x)}{\sigma_{\text{LN}\,i}(x) \cdot \sqrt{d}} \, , \quad n_{\text{ID}} : x \to x,$$

$$\mu_{\text{BN}}^j : x \to \frac{1}{B} \sum_{i=1}^{B} x_i^j, \quad \sigma_{\text{BN}}^j : x \to \sqrt{\frac{1}{B} \sum_{i=1}^{B} \left(x_i^j\right)^2 - \mu_{\text{BN}}^j(x)^2},$$

$$\mu_{\text{LN}\,i} : x \to \frac{1}{d} \sum_{j=1}^{d} x_i^j, \quad \sigma_{\text{LN}\,i} : x \to \frac{\|x_i - \mu_{\text{LN}\,i}(x)\|_2}{\sqrt{d}}$$

When using no normalization at all, the projection $\ell_2$ norm rapidly increases during the first 100 epochs and stabilizes at around $3 \cdot 10^6$ as shown in Figure 7. Despite this behaviour, using no normalization still performs reasonably well (67.4%). The $\ell_2$ normalization performs the best.

## H Training with smaller batch sizes

The results described in Section 4 were obtained using a batch size of 4096 split over 512 TPU cores. Due to its increased robustness, BYOL can also be trained using smaller batch sizes without significantly decreasing performance. Using the same linear evaluation setup, BYOL achieves 73.7% top-1 accuracy when trained over 1000 epochs with a batch size of 512 split over 64 TPU cores (approximately 4 days of training). For this setup, we reuse the same setting as in Section 3, but use a base learning rate of 0.4 (appropriately scaled by the batch size) and $\tau_{\text{base}} = 0.9995$ with the same weight decay coefficient of $1.5 \cdot 10^{-6}$.

## I Details on Equation 4 in Section 3.2

In this section we clarify why BYOL's update is related to Eq. 4 from Section 3.2,

$$\nabla_\theta \mathbb{E}\left[\left\|q^\star(z_\theta) - z'_\xi\right\|_2^2\right] = \nabla_\theta \mathbb{E}\left[\left\|\mathbb{E}\left[z'_\xi | z_\theta\right] - z'_\xi\right\|_2^2\right] = \nabla_\theta \mathbb{E}\left[\sum_i \text{Var}(z'_{\xi,i} | z_\theta)\right]. \quad (4)$$

Recall that $q^\star$ is defined as

$$q^\star \triangleq \underset{q}{\arg\min}\, \mathbb{E}\left[\left\|q(z_\theta) - z'_\xi\right\|_2^2\right], \quad \text{where} \quad q^\star(z_\theta) = \mathbb{E}\left[z'_\xi | z_\theta\right], \quad (3)$$

and implicitly depends on $\theta$ and $\xi$; therefore, it should be denoted as $q^\star(\theta, \xi)$ instead of just $q^\star$. For simplicity we write $q^\star(\theta, \xi)(z_\theta)$ as $q^\star(\theta, \xi, z_\theta)$ the output of the optimal predictor for any parameters $\theta$ and $\xi$ and input $z_\theta$.

BYOL updates its online parameters following the gradient of Eq. 4, but considering only the gradients of $q$ with respect to its third argument $z$ when applying the chain rule. If we rewrite

$$\mathbb{E}\left[\left\|q^\star(\theta, \xi, z_\theta) - z'_\xi\right\|_2^2\right] = \mathbb{E}\left[L(q^\star(\theta, \xi, z_\theta), z'_\xi)\right], \tag{11}$$

the gradient of this quantity w.r.t. $\theta$ is

$$\frac{\partial}{\partial \theta}\mathbb{E}\left[L(q^\star(\theta, \xi, z_\theta), z'_\xi)\right] = \mathbb{E}\left[\frac{\partial L}{\partial q} \cdot \frac{\partial q^\star}{\partial \theta} + \frac{\partial L}{\partial q} \cdot \frac{\partial q^\star}{\partial z} \cdot \frac{\partial z_\theta}{\partial \theta}\right], \tag{12}$$

where $\frac{\partial q^\star}{\partial \theta}$ and $\frac{\partial q^\star}{\partial z}$ are the gradients of $q^\star$ with respect to its first and last argument. Using the envelope theorem, and thanks to the optimality condition of the predictor, the term $\mathbb{E}\left[\frac{\partial L}{\partial q} \cdot \frac{\partial q^\star}{\partial \theta}\right] = 0$. Therefore, the remaining term $\mathbb{E}\left[\frac{\partial L}{\partial q} \cdot \frac{\partial q^\star}{\partial z} \cdot \frac{\partial z_\theta}{\partial \theta}\right]$ where gradients are only back-propagated through the predictor's input is exactly the direction followed by BYOL.

## J   Importance of a near-optimal predictor

In this part we build upon the intuitions of Section 3.2 on the importance of keeping the predictor near-optimal. Specifically, we show that it is possible to remove the exponential moving average in BYOL's target network (*i.e.*, simply copy weights of the online network into the target) without causing the representation to collapse, provided the predictor remains sufficiently good.

### J.1   Predictor learning rate

In this setup, we remove the exponential moving average (*i.e.*, set $\tau = 0$ over the full training in Eq. 1), and multiply the learning rate of the predictor by a constant $\lambda$ compared to the learning rate used for the rest of the network; all other hyperparameters are unchanged. As shown in Table 21, using sufficiently large values of $\lambda$ provides a reasonably good level of performance and the performance sharply decreases with $\lambda$ to $0.01\%$ top-1 accuracy (no better than random) for $\lambda = 0$.

To show that this effect is directly related to a change of behavior in the predictor, and not only to a change of learning rate in any subpart of the network, we perform a similar experiment by using a multiplier $\lambda$ on the *predictor*'s learning rate, and a different multiplier $\mu$ for the *projector*. In Table 22, we show that the representation typically collapses or performs poorly when the predictor learning rate is lower or equal to that of the projector. As mentioned in Section 3.2, we further hypothesize that one of the contributions of the target network is to maintain a near optimal predictor at all times.

### J.2   Optimal linear predictor in closed form

Similarly, we can get rid of the slowly moving target network if we use a closed form optimal predictor on the batch, instead of a learned, non-optimal one. In this case we restrict ourselves to a linear predictor,

$$q^\star = \operatorname*{arg\,min}_Q \left\|Z_\theta Q - Z'_\xi\right\|_2^2 = (Z_\theta^\mathsf{T} Z_\theta)^{-1} Z_\theta^\mathsf{T} Z'_\xi$$

with $\|\cdot\|_2$ being the Frobenius norm, $Z_\theta$ and $Z'_\xi$ of shape $(B, F)$ respectively the online and target projections, where $B$ is the batch size and $F$ the number of features; and $q^\star$ of shape $(F, F)$, the optimal linear predictor for the batch.

At 300 epochs, when using the closed form optimal predictor with a projection size of 16 (the predictor is a $16 \times 16$ matrix computed optimal on a batch of 4096 elements), and directly hard copying the weights of the online network to the target ($\tau = 0$), we obtain a top-1 accuracy of $50.3\%$.

| $\lambda$ | Top-1 |
|---|---|
| 0 | 0.01 |
| 1 | 5.5 |
| 2 | $62.8_{\pm1.5}$ |
| 10 | 66.6 |
| 20 | $66.3_{\pm0.3}$ |
| Baseline | 72.5 |

Table 21: Top-1 accuracy at 300 epochs when removing the slowly moving target network, directly hard copying the weights of the online network into the target network, and applying a multiplier to the predictor learning rate.

|  |  | $\lambda_{pred}$ | | | | |
|---|---|---|---|---|---|---|
|  |  | 1 | 1.5 | 2 | 5 | 10 |
| $\mu_{proj}$ | 1 | $3.2_{\pm2.9}$ | $25.7_{\pm6.6}$ | $60.8_{\pm2.9}$ | $66.7_{\pm0.4}$ | 66.9 |
|  | 1.5 | $1.4_{\pm2.0}$ | $9.2_{\pm7.0}$ | $55.2_{\pm5.8}$ | $61.5_{\pm0.6}$ | $66.0_{\pm0.3}$ |
|  | 2 | $2.0_{\pm2.8}$ | $5.3_{\pm1.9}$ | $15.8_{\pm13.4}$ | $60.9_{\pm0.8}$ | 66.3 |
|  | 5 | $1.5_{\pm0.9}$ | $2.5_{\pm1.5}$ | $2.5_{\pm1.4}$ | $20.5_{\pm2.0}$ | $60.5_{\pm0.6}$ |
|  | 10 | 0.1 | $2.1_{\pm0.3}$ | $1.9_{\pm0.8}$ | $2.8_{\pm0.4}$ | $8.3_{\pm6.8}$ |

Table 22: Top-1 accuracy at 300 epochs when removing the slowly moving target network, directly hard copying the weights of the online network in the target network, and applying a multiplier $\mu$ to the projector and $\lambda$ to the predictor learning rate. The predictor learning rate needs to be higher than the projector learning rate in order to successfully remove the

Figure 8: BYOL sketch summarizing the method by emphasizing the neural architecture.

# J  BYOL pseudo-code in JAX

## J.1  Hyper-parameters

```python
HPS = dict(
    max_steps=int(1000. * 1281167 / 4096),  # 1000 epochs
    batch_size=4096,
    mlp_hidden_size=4096,
    projection_size=256,
    base_target_ema=4e-3,
    optimizer_config=dict(
        optimizer_name='lars',
        beta=0.9,
        trust_coef=1e-3,
        weight_decay=1.5e-6,
        # As in SimCLR and official implementation of LARS, we exclude bias
        # and batchnorm weight from the Lars adaptation and weightdecay.
        exclude_bias_from_adaption=True),
    learning_rate_schedule=dict(
        # The learning rate is linearly increase up to
        # its base value * batchisze / 256 after warmup_steps
        # global steps and then anneal with a cosine schedule.
        base_learning_rate=0.2,
        warmup_steps=int(10. * 1281167 / 4096),
        anneal_schedule='cosine'),
    batchnorm_kwargs=dict(
        decay_rate=0.9,
        eps=1e-5),
    seed=1337,
    )
```

## J.2  Network definition

```python
def network(inputs):
  """Build the encoder, projector and predictor."""
  embedding = ResNet(name='encoder', configuration='ResNetV1_50x1')(inputs)
  proj_out = MLP(name='projector')(embedding)
  pred_out = MLP(name='predictor')(proj_out)
  return dict(projection=proj_out, prediction=pred_out)

class MLP(hk.Module):
  """Multi Layer Perceptron, with normalization."""

  def __init(self, name):
    super().__init__(name=name)

  def __call__(self, inputs):
    out = hk.Linear(output_size=HPS['mlp_hidden_size'])(inputs)
    out = hk.BatchNorm(**HPS['batchnorm_kwargs'])(out)
    out = jax.nn.relu(out)
    out = hk.Linear(output_size=HPS['projection_size'])(out)
    return out

# For simplicity, we omit BatchNorm related states.
# In the actual code, we use hk.transform_with_state. The corresponding
# net_init function outputs both a params and a state variable,
# with state containing the moving averages computed by BatchNorm
net_init, net_apply = hk.transform(network)
```

## J.3 Loss function

```python
def loss_fn(online_params, target_params, image_1, image_2):
  """Compute BYOL's loss function.

  Args:
    online_params: parameters of the online network (the loss is later
      differentiated with respect to the online parameters).
    target_params: parameters of the target network.
    image_1: first transformation of the input image.
    image_2: second transformation of the input image.

  Returns:
    BYOL's loss function.
  """

  online_network_out_1 = net_apply(params=online_params, inputs=image_1)
  online_network_out_2 = net_apply(params=online_params, inputs=image_2)
  target_network_out_1 = net_apply(params=target_params, inputs=image_1)
  target_network_out_2 = net_apply(params=target_params, inputs=image_2)

  def regression_loss(x, y):
    norm_x, norm_y = jnp.linalg.norm(x, axis=-1), jnp.linalg.norm(y, axis=-1)
    return -2. * jnp.mean(jnp.sum(x * y, axis=-1) / (norm_x * norm_y))

  # The stop_gradient is not necessary as we explicitly take the gradient with
  # respect to online parameters only. We leave it to indicate that gradients
  # are not backpropagated through the target network.
  loss = regression_loss(online_network_out_1['prediction'],
                         jax.lax.stop_gradient(target_network_out_2['projection']))
  loss += regression_loss(online_network_out_2['prediction'],
                         jax.lax.stop_gradient(target_network_out_1['projection']))
  return loss
```

## J.4 Training loop

```python
def main(dataset):
  """Main training loop."""

  rng = jax.random.PRNGKey(HPS['seed'])
  rng, rng_init = jax.random.split(rng, num=2)
  dataset = dataset.batch(HPS['batch_size'])
  dummy_input = dataset.next()
  byol_state = init(rng_init, dummy_input)

  for global_step in range(HPS['max_steps']):
    inputs = dataset.next()

    rng, rng1, rng2 = jax.random.split(rng, num=3)
    image_1 = simclr_augmentations(inputs, rng1, image_number=1)
    image_2 = simclr_augmentations(inputs, rng2, image_number=2)
    byol_state = update_fn(
        **byol_state,
        global_step=global_step,
        image_1=image_1,
        image_2=image_2)

  return byol_state['online_params']
```

## J.5 Update function

```python
optimizer = Optimizer(**HPS['optimizer_config'])

def update_fn(online_params, target_params, opt_state, global_step, image_1,
              image_2):
  """Update online and target parameters.

  Args:
    online_params: parameters of the online network (the loss is  differentiated
      with respect to the online parameters only).
    target_params: parameters of the target network.
    opt_state: state of the optimizer.
    global_step: current training step.
    image_1: first transformation of the input image.
    image_2: second transformation of the input image.

  Returns:
    Dict containing updated online parameters, target parameters and
    optimization state.
  """
  # update online network
  grad_fn = jax.grad(loss_fn, argnums=0)
  grads = grad_fn(online_params, target_params, image_1, image_2)
  lr = learning_rate(global_step, **HPS['learning_rate_schedule'])
  updates, opt_state = optimizer(lr).apply(grads, opt_state, online_params)
  online_params = optix.apply_updates(online_params, updates)

  # update target network
  tau = target_ema(global_step, base_ema=HPS['base_target_ema'])
  target_params = jax.tree_multimap(lambda x, y: x + (1 - tau) * (y - x),
                                    target_params, online_params)
  return dict(
      online_params=online_params,
      target_params=target_params,
      opt_state=opt_state)

def init(rng, dummy_input):
  """BYOL's state initialization.

  Args:
    rng: random number generator used to initialize parameters.
    dummy_input: a dummy image, used to compute intermediate outputs shapes.

  Returns:
    Dict containing initial online parameters, target parameters and
    optimization state.
  """
  online_params = net_init(rng, dummy_input)
  target_params = net_init(rng, dummy_input)
  opt_state = optimizer(0).init(online_params)
  return dict(
      online_params=online_params,
      target_params=target_params,
      opt_state=opt_state)
```

## Footnotes

[8] https://github.com/Philip-Bachman/amdim-public/blob/master/costs.py