[Reviews · NeurIPS 2020]

Review 1

Summary and Contributions: This paper proposes a new approach to self-supervised learning. There are two major differences with previous methods: removing the negative pairs, and the online-target branches. Both designs are well ablated. The final accuracy is very high, as 74.3 using R50.

Strengths: ++ This paper works on a very important problem of unsupervised pretraining. ++ The findings that no negative pairs are required during training is amazing. ++ The accuracy is very high, which improves previous best approach by about 1.3 points (Info Min).

Weaknesses: No good explanations about why positive pair alone is enough for pretraining are provided. I would further increase my ratings if a good explanation is given.

Correctness: seems sound

Clarity: well written

Relation to Prior Work: clear

Reproducibility: Yes

Additional Feedback:


Review 2

Summary and Contributions: I thank the authors for a very considerate rebuttal that adressed a few comments I made, the relation to prior work and the question about non-collapse. I have increased my score. This paper explores using a target network to perform contrastive learning without negative examples. This approach outperforms standard contrastive methods and is more robust to smaller batch sizes and image augmentations.

Strengths: The model achieves state of the art in unsupervised learning on ImageNet. Despite much recent progress in this field, the authors show that without negative samples models can become even better at learning from simple augmentations. The empirical evaluation is thorough and convincing.

Weaknesses: Why does this approach work instead of collapsing to a trivial solution? Have the authors explored any theoretical or practical justifications or experiments that help understanding why this approach works?

Correctness: The presented methods are theoretically and empirically very clean and raise no concerns. One nit: is there a specific reason why equation 2 and preceding sentences speak about normalizing and then measuring an L2 distances rather than just saying cosine similiarity?

Clarity: The paper is extremely well written and easy to follow.

Relation to Prior Work: There are two prior works that appear to be extremely similar: Laine & Aila (2016) Temporal Ensembling for Semi-Supervised Learning. Tarvainen & Valpola (2017) Mean teachers are better role models: Weight-averaged consistency targets improve semi-supervised deep learning results. Could the authors please comment why these are not cited and how this current work is not incremental on those former works.

Reproducibility: Yes

Additional Feedback: I will consider raising my score if the authors address the two main concerns: a) relation to prior work, b) (tentative) theoretical and or empirical explanation why this approach does not collapse. Why are the number in Tab. 2b lower than the corresponding entries in Tab. 1b? Should the semi-supervised setting not perform better? line 222: is there a verb missing? 224-228: are these changes relevant? 241-251: SimCLR is also maximing some cosine similarity between positive pairs. Can the authors be more explicit what the difference is here? footnote 4: this seems to be ignorring the effect of gradient noise that is larger for smaller batches? Tab5: given this table, how is tau chosen for the main results? Supp. material: 516-517: these lines appear again below? 559: is that 'local validation set' part of the 1% of data used for training? eq. 5: this is not the standard InfoNCE formulation, can you please derive this e.g. from the formulation in (Poole et al., 2019, On Variational Bounds of Mutual Information)? 745-756: this is going some way towards understanding why the approach works and does not collapse, maybe the authors could expand?


Review 3

Summary and Contributions: This paper proposes a new method for self-supervised learning, which doesn't rely on negative pairs which are required in most of the contrastive-based self-supervised learning techniques. Two networks are built, and the online model tries to predict the outputs of target model, which is exponentially averaged update by the online model's parameters, similar to mean-teacher and MoCo. Very good performances have been achieved by BYOL.

Strengths: -BYOL gets rid of negative pairs in contrastive learning, which eases the self-supervise learning problem. -The proposed method is very simple. And kind of too simple to be true. -Very comprehensive evaluation of BYOL, and in most the cases, BYOL outperforms the SOTA SimCLR and MoCo.

Weaknesses: -As mentioned in the paper, the proposed method has a trivial solution, that both models output 0's. Although empirically it doesn't, any theoretical support? -It doesn't say the code will be released. To me, the method is too simple to be true. I tried to reimplement it, but no success. It is highly recommend to opensource the code for reproduceable research. -The detection results are weird. Why so low? Frozen representation? How can you learn detection with frozen representation? Please use the standard settings, e.g. as in MoCo. The results are not convincing! Some questions: -L38, I understand negative pairs could be a limit of contrastive learning, e.g. batch-size. But why without negative pairs can improve robustness? -Why the online network needs a predictor? Actually, the predictor is just another embedding layer. Any ablation on removing the predictor from online model and adding a predictor to the target model? -Fig. 3(a), if the method is not coupled with batch size, why still have accuracy drop, especially from 256 to 128? For example, when you train classification network, the accuracies should be almost the same for 256 and 128. Isn't 128 enough for BN? Batch size of 32 should be enough to get reliable BN statistics. The explanation is not convincing.

Correctness: Seems mostly correct

Clarity: Yes, mostly

Relation to Prior Work: To me, (1) is more closely related to MoCo and Mean-teacher than [51]. It seems this paper tries to weaken the relations to MoCo and MT on purpose.

Reproducibility: No

Additional Feedback: The rebuttal has resolved most of my concerns. A few suggestions for the camera-ready. - release the code for reproduction, and show some results for shorter training, e.g. 200 epochs. Not everyone in the community has the resources to run experiments of 1000 epochs, especially in the university. - Footnote 4 is confusing and not the true reason. Please make it clear as in the rebuttal. - Fig. 5b shows some interesting ablations, but I missed most of them due to the lack of description. And they are not just ablation to contrastive methods, when \beta=0, they are simply on BYOL. Make them clear. -be honest to the relations to prior works, e.g. MoCo and Mean Teacher.

[Author Response · NeurIPS 2020]

We thank the reviewers for their very insightful and helpful comments and address their questions and remarks below.

**Relationship to Mean Teacher/Temporal Ensembling:** We are grateful to reviewers for pointing out this relevant related work that we missed in our original literature review. We will add the two citations in the related work section along with a paragraph on semi-supervised learning and connect MT and our current ablation study. Indeed, Table 5b (line 7: no predictor, $\beta = 0$, $0.2\%$) corresponds to using an MT-like approach in unsupervised learning (i.e., removing MT's classification loss) and we show that this approach does collapse. BYOL's novelty over MT is to perform well even without labels or classification loss, thanks to the addition of a predictor.

**Code and reproducibility:** We will release an open-source version of our full pretraining pipeline, the pretrained checkpoints, and the linear evaluation pipeline on ImageNet within the next two weeks. To further improve reproducibility and accessibility, we will also provide a single-GPU setup for pretraining on the smaller Imagenette dataset.

**Importance of a near-optimal predictor:** As the predictor is only applied to the online branch, its role is to make the architecture asymmetric rather than just making the network deeper. Table 5b already shows the importance of combining a predictor and a target network: the representation does collapse when either is removed. Furthermore, new experiments show that keeping the predictor near-optimal at all times is key to preventing collapse, which may be one of the roles of BYOL's target network. We further found that we can remove the target network without collapse by making the predictor near-optimal, either by (i) using an optimal *linear* predictor (obtained by linear regression on the current batch) before back-propagating the error through the network ($52.5\%$ top-1 accuracy at 300 epochs), or (ii) increasing the learning rate of the predictor ($66.5\%$ top-1). By contrast, increasing the learning rates of both projector *and* predictor (without target network) yields poor results ($\approx 25\%$ top-1).

**Explaining BYOL's non-collapse:** Similarly to GANs, BYOL uses two sets of parameters that are not minimizing the same objective. Thus, there is no *a priori* reason for BYOL's dynamics to converge to a global minimum of $||\bar{q}_\theta(z_\theta) - \bar{z}'_\theta||^2$, as they are not following the gradient of this loss ($\mathcal{L}_\theta^{\text{BYOL}}$ uses $\bar{z}'_\xi$). While these dynamics still admit undesirable equilibria where all images are mapped to the same constant projection (e.g., all zeros), BYOL's empirical performance seems to indicate that such equilibria may be unstable. We hypothesize that maintaining a near optimal predictor at all times is key to avoid collapsed solutions. When using an optimal predictor, BYOL minimizes the (expected) conditional variance of the target projection given the online projection . With a fixed target network, adding more information to the online projection can reduce this conditional variance, but cannot increase it. For example, training dynamics will always tend not to collapse features from the online network, as for any constant $C$ and variables $X$ and $Y$, $\text{Var}(Y|X) \leq \text{Var}(Y|C)$. More generally BYOL is encouraged to keep features from the online projection diverse by latching onto any source of variability $Z$ (stemming, e.g., from noise in training dynamics) distinct from existing features, as $\text{Var}(Y|X, Z) \leq \text{Var}(Y|X, h(X))$ for any variables $X, Y, Z$ and any function $h$. We will add these additional discussions and experiments to our submission to clarify the role of the predictor.

**Note on Fig 3a/footnote 4 (batch size):** When dividing the batch size by N, we also average gradients for N steps. Without batch-norm (for BYOL), the two computations would be exactly equivalent. With batch-norm for BYOL, the two computations only differ in how the batch-norm statistics are computed.

**Note on robustness:** As described in Section 5, contrastive methods need to make the discrimination task challenging, which requires many negative examples (large batch size) and absence of uninformative features that are easy to discriminate (strong transformations). They stop learning once their prediction is sufficiently similar to the positives compared to the negatives. Instead BYOL does not rely on comparing positive and negative pairs and keeps latching on new information thanks to the predictor. It should therefore not be as sensitive either to batch-size or transformations.

**Answers to Reviewer 1:** We thank you for your positive comments, and we hope to have answered your questions in explaining BYOL's non-collapse. As an optimal predictor seems sufficient to favor a diverse representation and stabilize the training, it removes the need for negative examples that are required to balance contrastive objectives.

**Answers to Reviewer 2:** We thank you for your in-depth questions and comments. Non-collapse and L.745-756: see discussion above. Table 1b vs 2b: Linear evaluation trains the linear layer on top of the representation with $100\%$ of the labels, while semi-supervised learning only uses 1 or $10\%$ of the dataset to finetune the full network (including the linear layer). This explains the difference in performance. Note that the same trend is observed in SimCLR and other related literature. L.224-228: these changes allow us to use the same hyperparameters as those optimized for SimCLR in their paper; these parameters are however not optimal at 1000 epochs, though the gap is low. L.241-251: the cosine similarity in BYOL is not between the same terms, and crucially, involves the output of the predictor. See also note on robustness above. $\ell^2$ vs cosine: we wanted to emphasize that BYOL also works well with just an $\ell^2$ loss (no normalization, as per table 19). Table 5: table 5 provides the trend of the target update, we further sweep on this hyperparameter in our main experiments. L.559: No, it is disjoint. Equation 5: eq. (5) is a generalization of the InfoNCE loss with added temperature parameter $\alpha$, in expanded form. We will add the derivation from the $I_{NCE}$ equation (10) of Poole et al. in Appendix F.4., to clarify differences with our equation (5).

**Answers to Reviewer 3:** Thank you for your insights and your time. Reproducibility: see discussion above. Detection results: we are using the same setup as in MoCo's Table 4 (fine-tune the representation on `trainval2007` only, not `trainval2007+12`), for which BYOL gains $+2.6 AP_{50}$ over MoCo. L34: see note on robustness above.

[Meta-Review · NeurIPS 2020]

This paper proposes a new method for self-supervised learning, which doesn't require negative pairs, unlike other contrastive approaches. It instead makes use of a target network. The reviewers unanimously voted to accept -- they really liked this paper and found it to be quite novel.